# *Plasmodium* P36 determines host cell receptor usage during sporozoite invasion

Giulia Manzoni[1], Carine Marinach[1†], Selma Topçu[1†], Sylvie Briquet[1], Morgane Grand[1], Matthieu Tolle[1], Marion Gransagne[1], Julien Lescar[1], Chiara Andolina[2,3], Jean-François Franetich[1], Mirjam B Zeisel[4,5], Thierry Huby[6], Eric Rubinstein[7,8], Georges Snounou[1], Dominique Mazier[1,9], François Nosten[2,3], Thomas F Baumert[4,5,10], Olivier Silvie[1*]

[1]Sorbonne Universités, UPMC Univ Paris 06, INSERM, CNRS, Centre d'Immunologie et des Maladies Infectieuses, U1135, ERL8255, Paris, France; [2]Shoklo Malaria Research Unit, Mahidol-Oxford Tropical Medicine Research Unit, Faculty of Tropical Medicine, Mahidol University, Mae Sot, Thailand; [3]Centre for Tropical Medicine and Global Health, Nuffield Department of Medicine, University of Oxford, Oxford, United Kingdom; [4]INSERM, U1110, Institut de Recherche sur les Maladies Virales et Hépatiques, Strasbourg, France; [5]Université de Strasbourg, Strasbourg, France; [6]Sorbonne Universités, UPMC Univ Paris 06, INSERM, Institute of Cardiometabolism and Nutrition, UMR_S 1166, Paris, France; [7]INSERM, U935, Villejuif, France; [8]Université Paris Sud, Institut André Lwoff, Villejuif, France; [9]Assistance Publique Hôpitaux de Paris, Centre Hospitalo-Universitaire Pitié-Salpêtrière, Paris, France; [10]Institut Hospitalo-Universitaire, Pôle Hépato-digestif, Hopitaux Universitaires de Strasbourg, Strasbourg, France

*For correspondence: olivier.silvie@inserm.fr

†These authors contributed equally to this work

**Abstract** *Plasmodium* sporozoites, the mosquito-transmitted forms of the malaria parasite, first infect the liver for an initial round of replication before the emergence of pathogenic blood stages. Sporozoites represent attractive targets for antimalarial preventive strategies, yet the mechanisms of parasite entry into hepatocytes remain poorly understood. Here we show that the two main species causing malaria in humans, *Plasmodium falciparum* and *Plasmodium vivax*, rely on two distinct host cell surface proteins, CD81 and the Scavenger Receptor BI (SR-BI), respectively, to infect hepatocytes. By contrast, CD81 and SR-BI fulfil redundant functions during infection by the rodent parasite *P. berghei*. Genetic analysis of sporozoite factors reveals the 6-cysteine domain protein P36 as a major parasite determinant of host cell receptor usage. Our data provide molecular insights into the invasion pathways used by different malaria parasites to infect hepatocytes, and establish a functional link between a sporozoite putative ligand and host cell receptors.

## Introduction

Hepatocytes are the main cellular component of the liver and the first replication niche for the malaria-causing parasite *Plasmodium*. Malaria begins with the inoculation of sporozoites into the host skin by infected *Anopheles* mosquitoes. Sporozoites rapidly migrate to the liver and actively invade hepatocytes by forming a specialized compartment, the parasitophorous vacuole (PV), where they differentiate into thousands of merozoites (*Ménard et al., 2013*). Once released in the blood, merozoites invade and multiply inside erythrocytes, causing the malaria disease.

**eLife digest** Malaria is an infectious disease that affects millions of people around the world and remains a major cause of death, especially in Africa. It is caused by *Plasmodium* parasites, which are transmitted by mosquitoes to mammals. Once in the mammal, the parasites infect liver cells, where they multiply. Previous studies have suggested that proteins on the surface of the liver cells and on the parasite affect how *Plasmodium* infects liver cells. Understanding how these proteins enable the parasites to enter the cells may help researchers to develop treatments that interrupt the parasite life cycle and prevent infection.

Manzoni et al. have now investigated how different malaria parasite species interact with liver cells. The main parasite species that infect humans are *Plasmodium falciparum* in Africa and *Plasmodium vivax* outside Africa. Manzoni et al. found that *P. falciparum* and *P. vivax* infect human liver cells by two different routes: *P. falciparum* interacts with a liver cell protein called CD81, and *P. vivax* interacts with a liver cell protein called SR-BI. Further experiments that used mutant forms of malaria parasites that infect mice showed that a parasite protein called P36 determines which liver cell protein the parasite will interact with.

The next step is to understand how P36 interacts with the liver cell proteins and to identify other parasite proteins that help *Plasmodium* to invade cells. In the future, such knowledge may help to develop a highly effective malaria vaccine.

Under natural transmission conditions, infection of the liver is an essential, initial and clinically silent phase of malaria, and therefore constitutes an ideal target for prophylactic intervention strategies. However, the molecular mechanisms underlying *Plasmodium* sporozoite entry into hepatocytes remain poorly understood. Highly sulphated proteoglycans in the liver sinusoids are known to bind the circumsporozoite protein, which covers the parasite surface, and contribute to the homing and activation of sporozoites (*Frevert et al., 1993*; *Coppi et al., 2007*). Subsequent molecular interactions leading to sporozoite entry into hepatocytes have not been identified yet. Several parasite proteins have been implicated, such as the thrombospondin related anonymous protein (TRAP) (*Matuschewski et al., 2002*), the apical membrane antigen 1 (AMA-1) (*Silvie et al., 2004*), or the 6-cysteine domain proteins P52 and P36 (*van Dijk et al., 2005*; *Ishino et al., 2005*; *van Schaijk et al., 2008*; *Kaushansky et al., 2015*; *Labaied et al., 2007*), however their role during sporozoite invasion remains unclear (*Bargieri et al., 2014*).

Our previous work highlighted the central role of the host tetraspanin CD81, one of the receptors for the hepatitis C virus (HCV) (*Pileri et al., 1998*), during *Plasmodium* liver infection (*Silvie et al., 2003*). CD81 is an essential host entry factor for human-infecting *P. falciparum* and rodent-infecting *P. yoelii* sporozoites (*Silvie et al., 2003*, *2006a*). CD81 acts at an early step of invasion, possibly by providing signals that trigger the secretion of rhoptries, a set of apical organelles involved in PV formation (*Risco-Castillo et al., 2014*). Whereas CD81 binds the HCV E2 envelope protein (*Pileri et al., 1998*), there is no evidence for such a direct interaction between CD81 and *Plasmodium* sporozoites (*Silvie et al., 2003*). Rather, we proposed that CD81 acts indirectly, possibly by regulating an as yet unidentified receptor for sporozoites within cholesterol-dependent tetraspanin-enriched microdomains (*Silvie et al., 2006b*; *Charrin et al., 2009a*). Intriguingly, the rodent malaria parasite *P. berghei* can infect cells lacking CD81 (*Silvie et al., 2003*, *2007*), however the molecular basis of this alternative entry pathway was until now totally unknown.

Another hepatocyte surface protein, the scavenger receptor BI (SR-BI), was shown to play a dual role during malaria liver infection, first in promoting parasite entry and subsequently its development inside hepatocytes (*Yalaoui et al., 2008a*; *Rodrigues et al., 2008*). However, the contribution of SR-BI during parasite entry is still unclear. SR-BI, which is also a HCV entry factor (*Scarselli et al., 2002*; *Bartosch et al., 2003*), binds high-density lipoproteins with high affinity and mediates selective cellular uptake of cholesteryl esters (*Acton et al., 1996*). Yalaoui *et al.* reported that SR-BI is involved indirectly during *P. yoelii* sporozoite invasion, by regulating the levels of membrane cholesterol and the expression of CD81 and its localization in tetraspanin-enriched microdomains (*Yalaoui et al., 2008a*). In another study, Rodrigues *et al.* observed a reduction of *P. berghei* invasion of Huh-7 cells

upon SR-BI inhibition (*Rodrigues et al., 2008*). Since CD81 is not required for *P. berghei* sporozoite entry into Huh-7 cells (*Silvie et al., 2007*), these results suggested a CD81-independent role for SR-BI. More recently, Foquet *et al.* showed that anti-CD81 but not anti-SR-BI antibodies inhibit *P. falciparum* sporozoite infection in humanized mice engrafted with human hepatocytes (*Foquet et al., 2015*), questioning the role of SR-BI during *P. falciparum* infection.

These conflicting results prompted us to revisit the contribution of SR-BI during *P. falciparum*, *P. yoelii* and *P. berghei* sporozoite infections. For the first time, we also explored the role of CD81 and SR-BI during hepatocyte infection by *P. vivax*, a widely distributed yet highly neglected cause of malaria in humans, for which the contribution of hepatocyte surface receptors has not been investigated to date.

Here, we show that SR-BI is an important entry factor for *P. vivax* but not for *P. falciparum* or *P. yoelii* sporozoites. Remarkably, SR-BI and CD81 fulfil redundant functions during host cell invasion by *P. berghei* sporozoites, which can use one or the other molecule. We further investigated parasite determinants associated with host cell receptor usage. We show that genetic depletion of P52 and P36, two members of the *Plasmodium* 6-cysteine domain protein family, abrogates sporozoite productive invasion and mimics the inhibition of CD81 and/or SR-BI entry pathways, in both *P. berghei* and *P. yoelii*. Finally, we identify P36 as the molecular driver of *P. berghei* sporozoite entry via SR-BI. Our data, by revealing a functional link between parasite and host cell entry factors, pave the way towards the identification of ligand-receptor interactions mediating *Plasmodium* infection of hepatocytes, and open novel perspectives for preventive and therapeutic approaches.

## Results

### Antibodies against SR-BI inhibit *P. vivax* but not *P. falciparum* or *P. yoelii* sporozoite infection

To evaluate the contribution of CD81 and SR-BI during *P. vivax* infection, we tested the effects of neutralizing CD81- and/or SR-BI- specific antibodies on *P. vivax* sporozoite infection in primary human hepatocyte cultures (*Mazier et al., 1984*). A monoclonal antibody (mAb) against the main extracellular domain of CD81, previously shown to inhibit *P. falciparum* sporozoite infection (*Silvie et al., 2003*), had no effect on the number of *P. vivax*-infected cells in vitro (*Figure 1A*). In sharp contrast, a mouse mAb specific for SR-BI (*Zahid et al., 2013*) greatly reduced infection, with no additive effect of anti-CD81 antibodies (*Figure 1A*). The same inhibitory effect was observed using polyclonal anti-SR-BI antibodies (*Figure 1B*). We performed the same experiments with *P. falciparum* sporozoites and found that anti-CD81 but not anti-SR-BI antibodies inhibit *P. falciparum* infection in vitro (*Figure 1C*), in agreement with the in vivo data from Foquet *et al* (*Foquet et al., 2015*). These data strongly suggest that *P. vivax* and *P. falciparum* sporozoites use distinct entry pathways to infect hepatocytes, reminiscent of the differences between *P. berghei* and *P. yoelii* (*Silvie et al., 2003*).

Two distinct populations of *P. vivax* exo-erythrocytic forms (EEFs) could be distinguished in the infected cultures, large EEFs representing replicating schizonts, and small EEFs that may correspond to hypnozoites (*Dembele et al., 2011*) (*Figure 1D*). Anti-SR-BI antibodies reduced the numbers of large and small EEFs to the same extent (*Figure 1E*), suggesting an effect on sporozoite invasion rather than on parasite intracellular development.

### SR-BI is required for productive invasion of CD81-null cells by *P. berghei* sporozoites

In order to investigate in more details the role of SR-BI during sporozoite entry, we used the more tractable rodent malaria parasite *P. berghei*. Indeed, *P. berghei* sporozoites readily infect HepG2 cells, which lack CD81 (*Charrin et al., 2001*; *Berditchevski et al., 1996*) but express high levels of SR-BI (*Figure 2—figure supplement 1*), raising the possibility that this rodent parasite uses a SR-BI route to infect CD81-null cells. To test this hypothesis, *P. berghei* sporozoites constitutively expressing GFP (PbGFP) (*Manzoni et al., 2014*) were incubated with HepG2 cells in the presence of increasing concentrations of polyclonal anti-SR-BI rabbit antibodies. We observed a dramatic and dose-dependent reduction of the number of EEF-infected cells induced by anti-SR-BI antibodies (*Figure 2A*). Quantification of host cell invasion by FACS demonstrated that the rabbit anti-SR-BI

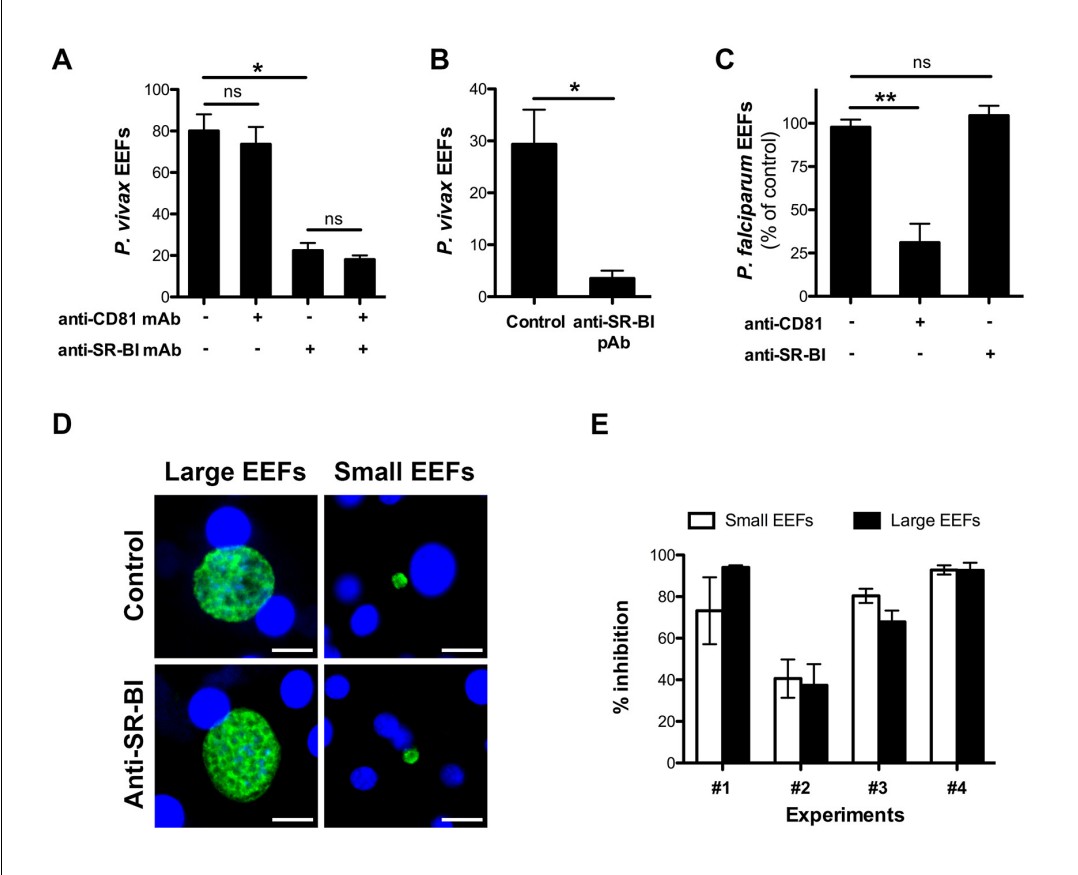

**Figure 1.** Anti-SR-BI antibodies inhibit *P. vivax* but not *P. falciparum* sporozoite infection. (A) Primary human hepatocyte cultures were incubated with *P. vivax* sporozoites in the presence of anti-CD81 and/or anti-SR-BI mAbs at 20 μg/ml, and the number of EEF-infected cells was determined 5 days post-infection after labeling of the parasites with anti-HSP70 antibodies. (B) Primary human hepatocytes were incubated with *P. vivax* sporozoites in the presence or absence of anti-SR-BI polyclonal rabbit serum (diluted 1/100), and the number of EEFs was determined at day 5 by immunofluorescence. (C) Primary human hepatocyte cultures were incubated with *P. falciparum* sporozoites in the presence of anti-CD81 mAb (20 μg/ml) and/or anti-SR-BI polyclonal rabbit serum (diluted 1/100), and the number of EEF-infected cells was determined 5 days post-infection after labeling of the parasites with anti-HSP70 antibodies. Results from three independent experiments are shown and expressed as the percentage of control (mean ±SD). (D) Immunofluorescence analysis of *P. vivax* EEFs at day 5 post-infection of primary human hepatocytes. Parasites were labeled with anti-HSP70 antibodies (green), and nuclei were stained with Hoechst 33342 (blue). Large EEFs and small EEFs were observed in both control and anti-SR-BI antibody-treated cultures. Scale bars, 10 μm. (E) Inhibitory activity of anti-SR-BI antibodies on small EEFs (white histograms) and large EEFs (black histograms). The results from four independent experiments are shown, and are expressed as the percentage of inhibition observed with anti-SR-BI antibodies as compared to the control.

antibodies block infection at the invasion step (*Figure 2B*). Similar results were obtained with polyclonal rat antibodies and a mouse mAb directed against human SR-BI (*Figure 2—figure supplement 2*).

As a complementary approach, we used small interfering RNA (siRNA) to specifically knockdown SR-BI expression in HepG2 cells (*Figure 2D*). SR-BI silencing caused a dramatic reduction of the number of EEF-infected cells (*Figure 2E*), but had no significant effect on the intracellular development of the few invaded parasites (*Figure 2F*). *Plasmodium* sporozoites migrate through several cells before invading a final one inside a PV (*Mota et al., 2001*). Sporozoite cell traversal was increased in SR-BI-depleted HepG2 cells, as compared to the control (*Figure 2—figure supplement 3*). This is likely due to the robust cell traversal activity of *P. berghei* sporozoites, which continue to migrate through cells when productive invasion is impaired. Altogether, these data establish that SR-BI is a major entry factor for *P. berghei* sporozoites in CD81-null HepG2 cells.

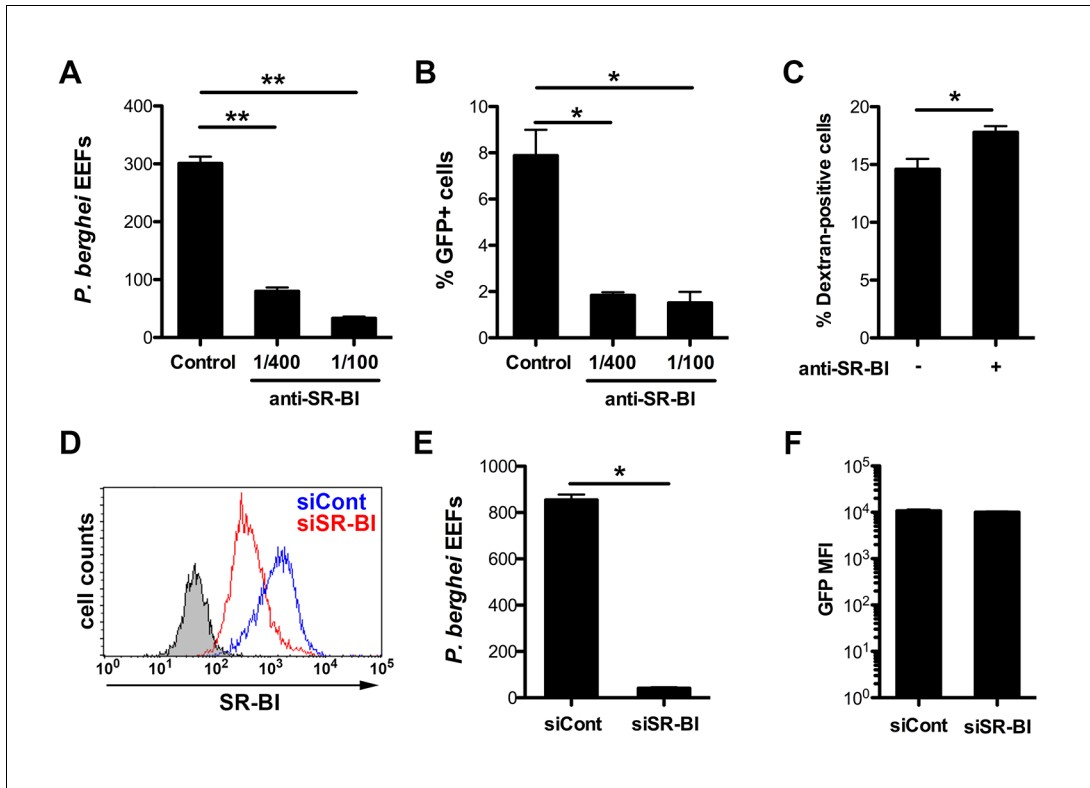

**Figure 2.** Infection of human HepG2 cells by *P. berghei* sporozoites depends on SR-BI. (**A**) HepG2 cells were incubated with *P. berghei* sporozoites for 3 hr in the absence (Control) or presence of increasing concentrations of rabbit polyclonal SR-BI antisera. Infected cultures were further incubated for 24 hr before quantification of EEFs-infected cells by fluorescence microscopy. (**B**) HepG2 cell cultures were infected with PbGFP sporozoites as in A, and the number of invaded cells (GFP+) was quantified by FACS 3 hr post-infection. (**C**) HepG2 cells were incubated for 3 hr with PbGFP sporozoites and rhodamine-labeled dextran, in the presence or absence of anti-SR-BI antibodies. The percentage of traversed (dextran-positive) cells was then determined by FACS. (**D**) HepG2 cells transfected with siRNA oligonucleotides targeting SR-BI (siSR-BI, red histogram) or with a control siRNA (siCont, blue histogram) were stained with anti-SR-BI antibodies and analyzed by flow cytometry. The negative staining control is in grey. (**E**) *P. berghei* EEF number in HepG2 cells transfected with siRNA oligonucleotides targeting SR-BI (siSR-BI) or a control siRNA (siCont). (**F**) HepG2 cells transfected with siRNA oligonucleotides targeting SR-BI (siSR-BI) or a control siRNA (siCont) were infected with PbGFP sporozoites and incubated for 24 hr, before measurement of the mean fluorescence intensity (GFP MFI) of infected (GFP-positive) cells by FACS.

The following figure supplements are available for figure 2:

**Figure supplement 1.** CD81 and SR-BI surface expression in HepG2 and HepG2/CD81 cells.

**Figure supplement 2.** Anti-SR-BI antibodies neutralize *P. berghei* infection of HepG2 cells.

**Figure supplement 3.** Effect of SR-BI silencing on sporozoite cell traversal and invasion.

## CD81 and SR-BI play redundant roles during *P. berghei* sporozoite invasion

We next tested whether the presence of CD81 would affect SR-BI function during *P. berghei* sporozoite infection. We monitored invasion and replication of *P. berghei* sporozoites in HepG2 cells genetically engineered to express CD81 (HepG2/CD81) (*Figure 2—figure supplement 1*) (*Silvie et al., 2006a*), in the presence of anti-SR-BI and/or anti-CD81 antibodies. Strikingly, unlike in CD81-null HepG2 cells, anti-SR-BI antibodies had no inhibitory effect on *P. berghei* infection in

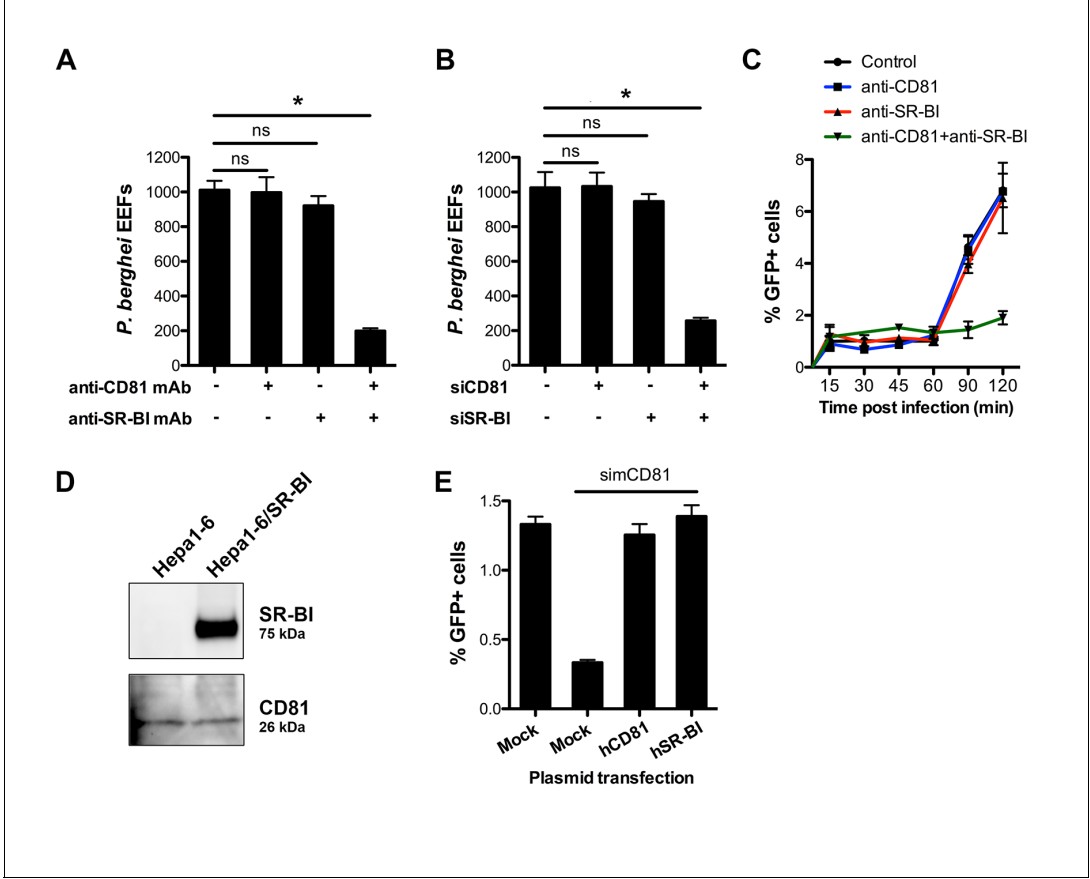

**Figure 3.** CD81 and SR-BI define alternative entry routes for *P. berghei* sporozoites. (**A**) HepG2/CD81 cells were incubated with *P. berghei* sporozoites in the presence or absence of anti-human CD81 and/or SR-BI mAbs, and the number of EEFs-infected cells was determined by fluorescence microscopy 24 hr post-infection. (**B**) *P. berghei* EEF numbers in HepG2/CD81 cells transfected with siRNA oligonucleotides targeting CD81 (siCD81) and/or SR-BI (siSR-BI). (**C**) HepG2/CD81 cells were incubated with *Pb*GFP sporozoites for 15–120 min, in the presence or absence of anti-CD81 and/or anti-SR-BI antibodies, then trypsinized and directly analyzed by FACS to quantify invaded (GFP-positive) cells. (**D**) Protein extracts from Hepa1-6 cells and Hepa1-6 cells transiently transfected with a human SR-BI expression plasmid were analyzed by Western blot using antibodies recognizing mouse and human SR-BI (Abcam) or mouse CD81 (MT81). (**E**) Hepa1-6 cells were transfected first with siRNA oligonucleotides targeting endogenous mouse CD81 (simCD81), then with plasmids encoding human CD81 (hCD81) or SR-BI (hSR-BI). Cells were then incubated with PbGFP sporozoites, and the number of infected (GFP-positive) cells was determined 24 hr post-infection by FACS.

The following figure supplement is available for figure 3:

**Figure supplement 1.** Effect of anti-CD81 and anti-SR-BI antibodies on *P. berghei* sporozoite cell traversal and intracellular development.

HepG2/CD81 cells (*Figure 3A*). Remarkably, whilst neither anti-SR-BI nor anti-CD81 antibodies alone had any significant impact on invasion (*Figure 3A*) or parasite intracellular development (*Figure 3—figure supplement 1*), the combination of CD81 and SR-BI antibodies markedly reduced the number of infected cells (*Figure 3A*). Similarly, siRNA-mediated silencing of either CD81 or SR-BI alone had no effect on infection, whereas simultaneous silencing of both factors greatly reduced infection (*Figure 3B*).

Blocking both CD81 and SR-BI was associated with an increase in cell traversal activity (*Figure 3—figure supplement 1*), suggesting that the concomitant neutralization of the two host factors prevented commitment to productive invasion. To directly test this hypothesis, we analysed the invasion kinetics of PbGFP sporozoites in HepG2/CD81 cells, in the presence of anti-SR-BI and/or anti-CD81 neutralizing antibodies. We have shown that in vitro sporozoite invasion follows a two-step kinetics (*Risco-Castillo et al., 2015*), with initially low invasion rates at early time points, reflecting cell

traversal activity, followed by a second phase of productive invasion associated with PV formation. In HepG2/CD81 cells, the invasion kinetics of *P. berghei* sporozoites in the presence of anti-SR-BI or anti-CD81 specific antibodies were comparable to those of the control without antibody (*Figure 3C*). In sharp contrast, blocking both CD81 and SR-BI simultaneously suppressed the second phase of productive invasion (*Figure 3C*). Based on these results, we conclude that SR-BI and CD81 are involved in the commitment to productive entry.

*P. berghei* sporozoites infect mouse hepatocytic Hepa1-6 cells via a CD81-dependent pathway, as shown by efficient inhibition of infection by CD81 specific antibodies or siRNA (*Silvie et al., 2007*). Interestingly, we failed to detect SR-BI in Hepa1-6 cells (*Figure 3D*), providing a plausible explanation as to why CD81 is required for *P. berghei* infection in this model. We tested whether ectopic expression of SR-BI would rescue *P. berghei* infection of Hepa1-6 cells upon silencing of endogenous murine CD81 by siRNA. CD81-silenced Hepa1-6 cells were transiently transfected with plasmids encoding human SR-BI or CD81, before exposure to *P. berghei* sporozoites. The number of infected cells was greatly reduced in CD81-silenced cells as compared to control non-silenced cells (*Figure 3E*), in agreement with our previous observations (*Silvie et al., 2007*). Remarkably, transfection of either human CD81 or human SR-BI was sufficient to rescue infection in CD81-silenced cells (*Figure 3E*), demonstrating that the two receptors can independently perform the same function to support *P. berghei* infection. Collectively, our data provide direct evidence that CD81 and SR-BI play redundant roles during productive invasion of hepatocytic cells by *P. berghei* sporozoites.

## *P. yoelii* sporozoites require host CD81 but not SR-BI for infection

We next investigated the contribution of SR-BI to *P. yoelii* sporozoite infection in HepG2/CD81 cells. *P. yoelii* infection was dramatically reduced by anti-CD81 antibodies, consistent with our previous work (*Silvie et al., 2006a*), but strikingly was not affected by anti-SR-BI antibodies (*Figure 4A*). In addition, *P. yoelii* infection was not affected by siRNA-mediated silencing of SR-BI, but was almost abolished upon knockdown of CD81 (*Figure 4B*). These results indicate that CD81 but not SR-BI plays a central role during *P. yoelii* sporozoite invasion in HepG2/CD81 cells, similarly to *P. falciparum* in human hepatocytes.

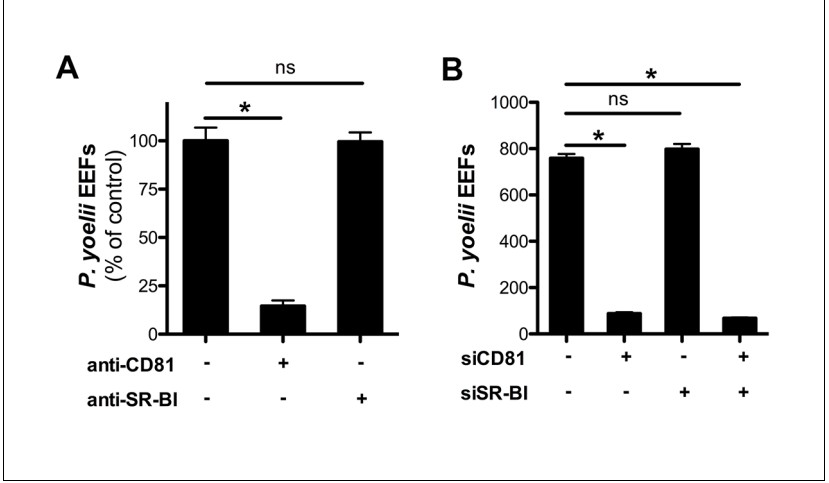

**Figure 4.** Infection of HepG2/CD81 cells by *P. yoelii* sporozoites depends on CD81 but not SR-BI. (**A**) HepG2/CD81 cells were incubated with *P. yoelii* sporozoites in the presence of anti-CD81 mAb (20 µg/ml) or anti-SR-BI polyclonal rabbit serum (diluted 1/100), and the number of EEF-infected cells was determined 24 hr post-infection by fluorescence microscopy. Results from three independent experiments are shown and expressed as the percentage of control (mean ± SD). (**B**) *P. yoelii* EEF numbers in HepG2/CD81 cells transfected with siRNA oligonucleotides targeting CD81 (siCD81) and/or SR-BI (siSR-BI).

## The 6-cysteine domain proteins P52 and P36 are required for sporozoite productive invasion irrespective of the host cell entry pathway

The data above show that *P. berghei* sporozoites can use either CD81 or SR-BI to infect hepatocytic cells, whereas *P. yoelii* utilizes CD81 but not SR-BI, suggesting that one or several *P. berghei* factors may be specifically associated with SR-BI usage. Among potential candidate parasite factors involved in sporozoite entry, we focused on the 6-cysteine domain proteins P52 and P36, two micronemal proteins of unknown function previously implicated during liver infection (*van Dijk et al., 2005*; *Ishino et al., 2005*; *van Schaijk et al., 2008*; *Kaushansky et al., 2015*; *Labaied et al., 2007*; *VanBuskirk et al., 2009*; *Mikolajczak et al., 2014*). To facilitate monitoring of the role of P52 and P36 during host cell invasion, we used a Gene Out Marker Out (GOMO) strategy (*Manzoni et al., 2014*) to generate highly fluorescent *p52/p36*-knockout parasite lines in *P. berghei* (*Figure 5—figure supplement 1*) and *P. yoelii* (*Figure 5—figure supplement 2*). Pure populations of GFP-expressing, drug selectable marker-free Pb*Δp52/p36* and Py*Δp52/p36* blood stage parasites were obtained and transmitted to mosquitoes in order to produce sporozoites.

Analysis of the kinetics of Pb*Δp52/p36* sporozoite invasion by FACS, in comparison to control PbGFP sporozoites, revealed that genetic ablation of *p52/p36* abrogates sporozoite productive invasion of HepG2 (SR-BI-dependent entry pathway) and Hepa1-6 cells (CD81-dependent entry pathway) (*Figure 5A and B*). Pb*Δp52/p36* sporozoite invasion followed similar kinetics to those observed for PbGFP sporozoites upon blockage of SR-BI or CD81, respectively, and was not modified by addition of anti-SRBI or anti-CD81 antibodies. Using antibodies specific for UIS4, a marker of the PV membrane (PVM) that specifically labels productive vacuoles (*Risco-Castillo et al., 2015*; *Mueller et al., 2005*), we confirmed that PbGFP but not Pb*Δp52/p36* parasites could form replicative PVs, in both HepG2 and HepG2/CD81 cells (*Figure 5C*). In both cell types, only very low numbers of EEFs were observed with Pb*Δp52/p36* parasites (*Figure 5D*), all of which were seemingly intranuclear and lacked a UIS4-labeled PVM (*Figure 5E*). We have shown before that intranuclear EEFs result from cell traversal events (*Silvie et al., 2006a*). Altogether these data demonstrate that Pb*Δp52/p36* sporozoites fail to productively invade host cells, irrespective of the entry route. Similar results were obtained with a *P. berghei Δp36* single knockout line using the GOMO strategy (*Figure 5—figure supplement 3*). Pb*Δp36* sporozoites did not invade HepG2 cells or HepG2/CD81 cells, reproducing a similar phenotype as PbGFP parasites in the presence of anti-CD81 and anti-SR-BI neutralizing antibodies (*Figure 5—figure supplement 4*).

We then examined the kinetics of *P. yoelii Δp52/p36* sporozoite invasion, in comparison to those of PyGFP sporozoites, in HepG2/CD81 versus HepG2 cells. Py*Δp52/p36* sporozoites showed a lack of productive invasion in HepG2/CD81 cells, reproducing the invasion kinetics of PyGFP in the CD81-null HepG2 cells (*Figure 5F*). The Py*Δp52/p36* mutant failed to form PV in HepG2/CD81 cells (*Figure 5G*), where only intranuclear EEFs lacking a UIS4-labeled PVM were observed, similarly to the control PyGFP parasites in HepG2 cells (*Figure 5H and I*).

Altogether, these results reveal that P52 and P36 are required for sporozoite productive invasion, in both *P. berghei* and *P. yoelii*, irrespective of the entry route.

## P36 is a key determinant of host cell receptor usage during sporozoite invasion

We further sought to investigate whether P52 and/or P36 proteins contribute to the selective usage of host cell receptors by different sporozoite species. We designed a trans-species genetic complementation strategy in which copies of *P. berghei* (Pb), *P. yoelii* (Py), *P. falciparum* (Pf) or *P. vivax* (Pv) P52 and P36 were introduced in the *Δp52/p36* parasites. For this purpose, we used centromeric plasmid constructs for stable expression of the transgenes from episomes (*Iwanaga et al., 2010*). Complementing Pb*Δp52/p36* sporozoites with PbP52 and PbP36 restored sporozoite infectivity to both HepG2 and HepG2/CD81 cells (*Figure 6A*), where the parasite formed UIS4-positive vacuoles (*Figure 6B, C and D*), confirming that genetic complementation was efficient. Remarkably, complementation of Pb*Δp52/p36* with PyP52 and PyP36 restored infection in HepG2/CD81 but not in HepG2 cells (*Figure 6A*), where only low numbers of UIS4-negative intranuclear EEFs were observed (*Figure 6B and D*). Thus, the concomitant replacement of PbP52 and PbP36 by their *P. yoelii* counterparts reproduced a *P. yoelii*-like invasion phenotype in chimeric *P. berghei* sporozoites, indicating

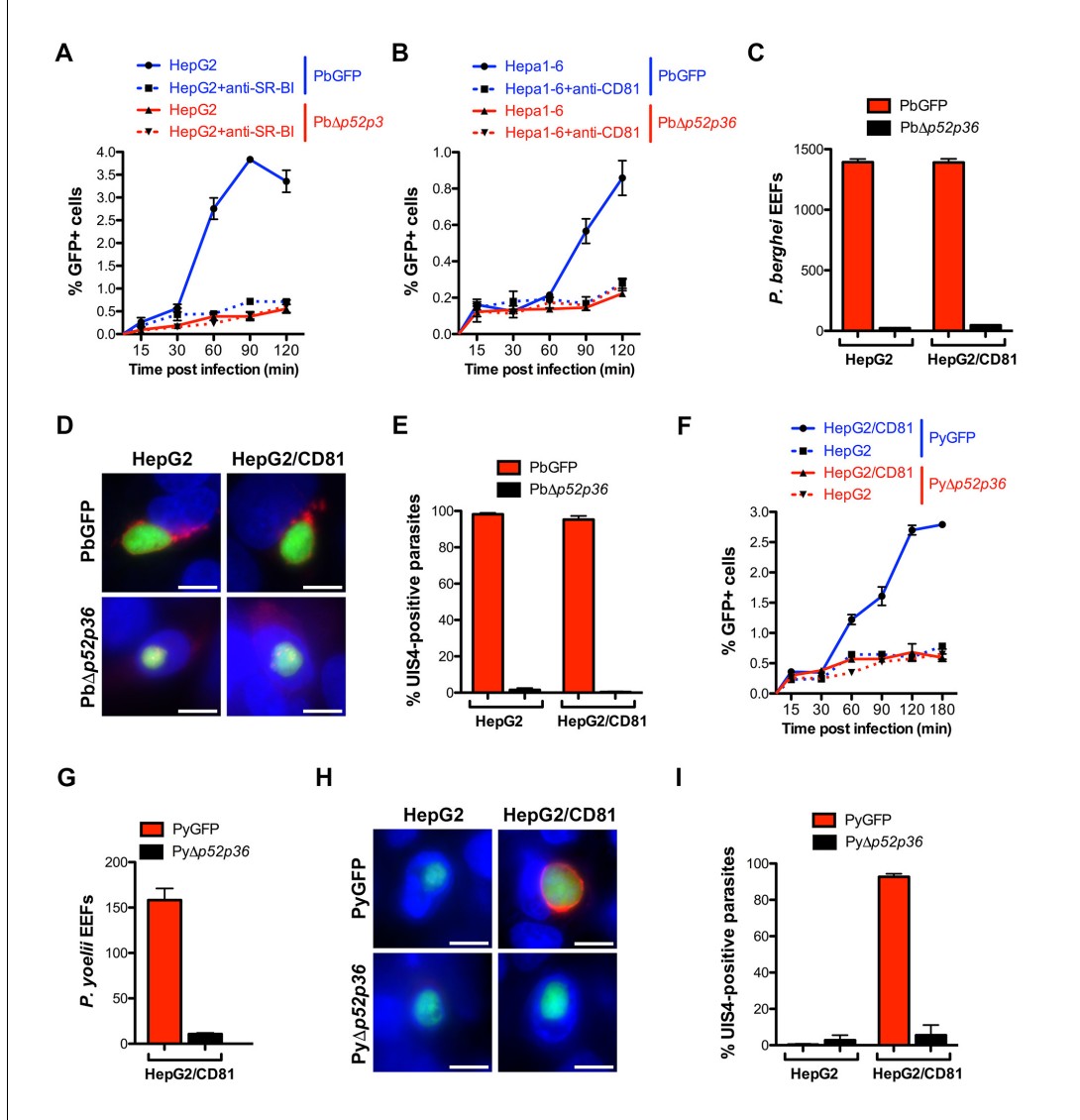

**Figure 5.** The 6-cys proteins P52 and P36 are required for productive host cell invasion. (A–B) HepG2 (A) or Hepa1-6 cells (B) were incubated with PbGFP (blue lines) or PbΔ*p52/p36* sporozoites (red lines) for 15–120 min, in the presence (dotted lines) or absence (solid lines) of anti-SR-BI (A) or anti-CD81 (B) antibodies. Cells were then trypsinized and directly analyzed by FACS to quantify invaded (GFP-positive) cells. (C) HepG2 and HepG2/CD81 cells were infected with PbGFP or PbΔ*p52/p36* sporozoites and the number of EEFs was determined 28 hr post-infection by fluorescence microscopy. (D) HepG2 and HepG2/CD81 cells infected with PbGFP or PbΔ*p52/p36* sporozoites were fixed at 28 hr post-infection, stained with anti-UIS4 antibodies (red) and the nuclear stain Hoechst 33342 (blue), and examined by fluorescence microscopy. Parasites were detected based on GFP expression (green). Scale bars, 10 μm. (E) Quantification of UIS4 expression in HepG2 and HepG2/CD81 cells infected with PbGFP (red) or PbΔ*p52/p36* (black). (F) HepG2 (dotted lines) and HepG2/CD81 cells (solid lines) were incubated with PyGFP (blue lines) or PyΔ*p52/p36* sporozoites (red lines) for 15–180 min, trypsinized, and directly analyzed by FACS to quantify invaded (GFP-positive) cells. (G) HepG2/CD81 cells were infected with PyGFP or PyΔ*p52/p36* sporozoites and the number of EEFs was determined 24 hr post-infection by fluorescence microscopy. (H) HepG2 and HepG2/CD81 cells infected with PyGFP or PyΔ*p52/p36* sporozoites were fixed at 24 hr post-infection, stained with anti-UIS4 antibodies (red) and the nuclear stain Hoechst 33342 (blue), and examined by fluorescence microscopy. Parasites were detected based on GFP expression (green). Scale bars, 10 μm. (I) Quantification of UIS4 expression in HepG2 and HepG2/CD81 cells infected with PyGFP (red) or PyΔ*p52/p36* (black).

The following figure supplements are available for figure 5:

**Figure supplement 1.** Targeted gene deletion of p52 and p36 in *P. berghei*.

**Figure supplement 2.** Targeted gene deletion of p52 and p36 in *P. yoelii*.

*Figure 5 continued on next page*

*Figure 5 continued*

**Figure supplement 3.** Targeted gene deletion of *p36* in *P. berghei*.

**Figure supplement 4.** P36 is required for *P. berghei* sporozoite entry via both SR-BI- and CD81-dependent routes.

that P52 and/or P36 contribute to the selective usage of a CD81-independent entry pathway in *P. berghei* sporozoites. Complementation of PbΔ*p52/p36* parasites with either *P. falciparum* or *P. vivax* P52 and P36 coding sequences did not restore infectivity of transgenic sporozoites, not only in HepG2 and HepG2/CD81 cells (*Figure 6—figure supplement 1A*), but also in primary human hepatocytes, the most permissive cellular system for human malaria sporozoites in vitro (*Figure 6—figure supplement 1B*). Hence it was not possible using this approach to assess the function of *P. falciparum* or *P. vivax* P52 and P36 in transgenic *P. berghei* sporozoites.

We next dissected the individual contribution of P52 and P36 by complementing PbΔ*p52/p36* parasites with mixed combinations of either PyP52 and PbP36 or PbP52 and PyP36. This approach revealed that P36 determines the ability of *P. berghei* sporozoites to enter cells via a CD81-independent route. Indeed, PbΔ*p52/p36* complemented with PyP52 and PbP36 infected both HepG2/CD81 and HepG2 cells (*Figure 6A*), forming UIS4-labeled PVs in both cell types (*Figure 6B, C and D*). P52 therefore is not responsible for the phenotypic difference between *P. berghei* and *P. yoelii*. In sharp contrast, complementation of PbΔ*p52/p36* with PbP52 and PyP36 restored sporozoite infectivity to HepG2/CD81 but not HepG2 cells, thus reproducing a *P. yoelii*-like invasion phenotype (*Figure 6A, B and D*).

In reciprocal experiments, we analysed whether expression of PbP36 would be sufficient to allow *P. yoelii* sporozoites to invade CD81-null cells. For this purpose, we performed genetic complementation experiments in PyΔ*p52/p36* parasites, using the same constructs employed with the *P. berghei* mutant. Strikingly, whilst HepG2 cells are normally refractory to *P. yoelii* productive invasion, complementation with *P. berghei* P52 and P36 protein was sufficient to confer chimeric *P. yoelii* mutants the capacity to infect HepG2 cells (*Figure 7A*). Most importantly, PyΔ*p52/p36* parasites complemented with PyP52 and PbP36 infected both HepG2 and HepG2/CD81 cells (*Figure 7A*). In particular, PyΔ*p52/p36* parasites expressing PbP36 became capable of forming UIS4-positive PVs in HepG2 cells (*Figure 7B*). Thus, the transgenic expression of PbP36 appears to be sufficient to recapitulate a *P. berghei*-like invasion phenotype in *P. yoelii* sporozoites. In contrast, complementation of PyΔ*p52/p36* parasites with PbP52 and PyP36 restored sporozoite infectivity in HepG2/CD81 cells only, but not in HepG2 cells (*Figure 7A and B*). This confirms that P52, although essential for sporozoite entry, is not directly associated with host receptor usage. Finally, invasion of PbP36-expressing PyΔ*p52/p36* sporozoites was abrogated by anti-SR-BI antibodies in HepG2 cells (*Figure 7C*), demonstrating that *P. berghei* P36 is a key determinant of CD81-independent entry via SR-BI.

## Discussion

Until now, the nature of the molecular interactions mediating *Plasmodium* sporozoite invasion of hepatocytes has remained elusive. Previous studies have identified CD81 and SR-BI as important host factors for infection of hepatocytes (*Silvie et al., 2003*; *Yalaoui et al., 2008a*; *Rodrigues et al., 2008*). Still, the relation between CD81 and SR-BI and their contribution to parasite entry was unclear (*Silvie et al., 2007*; *Foquet et al., 2015*), and, importantly, parasite factors associated with CD81 or SR-BI usage had not been identified. Here we demonstrate that CD81 and SR-BI define independent entry pathways for sporozoites, and identify the parasite protein P36 as a critical parasite factor that determines host receptor usage during hepatocyte infection.

Our data provide molecular insights into the host entry pathways used by different sporozoite species (*Figure 8A*). We show that *P. falciparum*, like *P. yoelii*, relies on CD81 but not SR-BI, in agreement with the recent observation that antibodies against CD81 but not anti-SR-BI induce protection in humanized mice infected with *P. falciparum* (*Foquet et al., 2015*). Our results are also consistent with the study from Yalaoui *et al.* showing that in primary mouse hepatocytes antibodies against SR-BI do not inhibit *P. yoelii* infection when co-incubated together with sporozoites (*Yalaoui et al., 2008a*). In the same study, the authors proposed a model where SR-BI indirectly

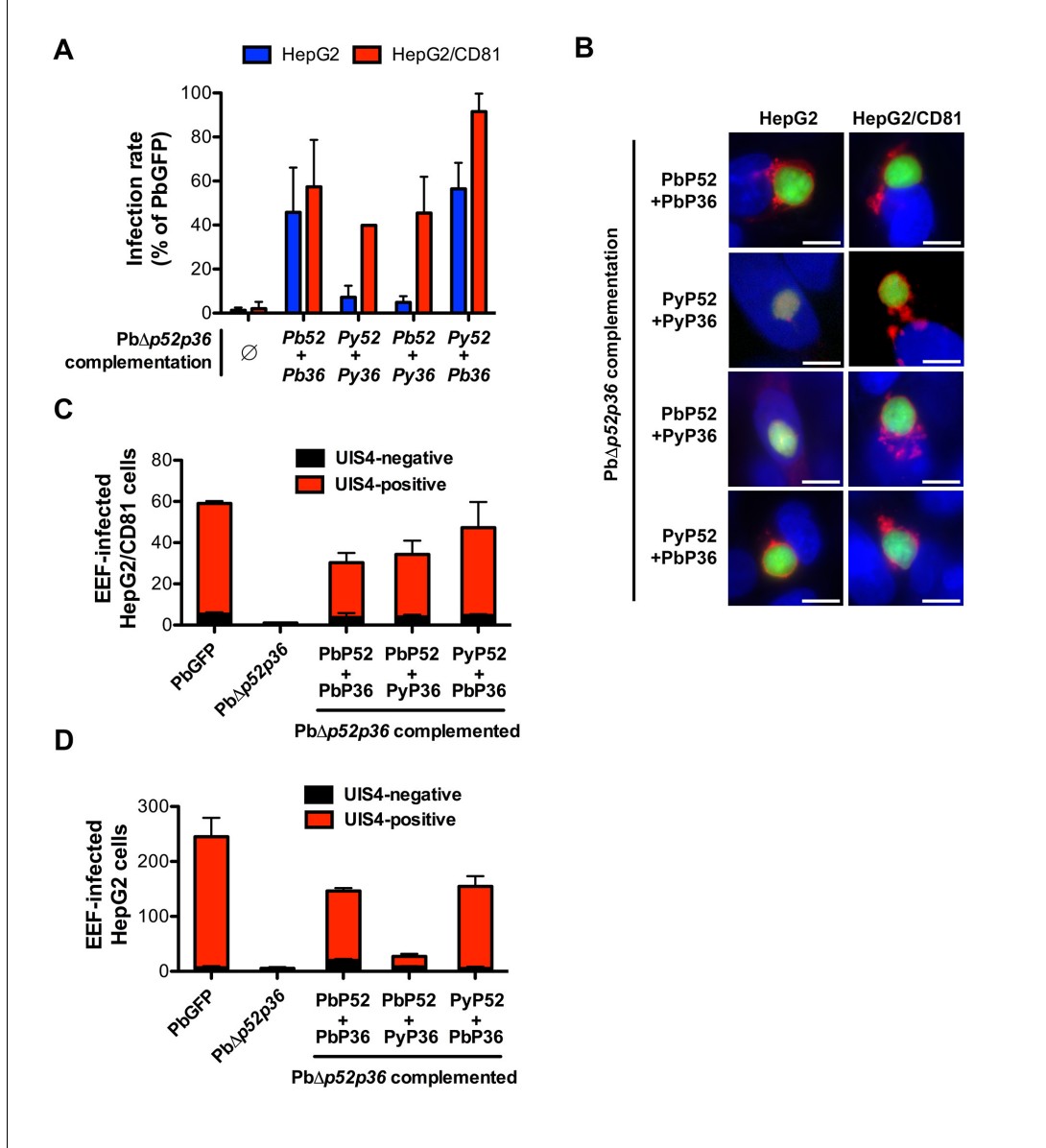

**Figure 6.** P36 mediates CD81-independent entry in *P. berghei* sporozoites. (**A**) HepG2 (blue histograms) or HepG2/CD81 cells (red histograms) were incubated with sporozoites from Pb*Δp52/p36* parasites genetically complemented with *P. berghei* and/or *P. yoelii* P52 and P36, and analysed by FACS or fluorescence microscopy to determine the number of GFP-positive cells 24 hr post-infection. Results from three independent experiments are shown and are expressed as the percentage of infection in comparison to control PbGFP-infected cultures (mean ±SD). (**B**) Immunofluorescence analysis of UIS4 expression in HepG2 or HepG2/CD81 cells infected with genetically complemented Pb*Δp52/p36* sporozoites. Cells were fixed with PFA 28 hr post-infection, permeabilized, and stained with anti-UIS4 antibodies (red) and the nuclear stain Hoechst 33342 (blue). Parasites were detected based on GFP expression (green). Scale bars, 10 μm. (**C–D**) HepG2/CD81 (**C**) and HepG2 (**D**) cells were infected with PbGFP, Pb*Δp52/p36* and complemented Pb*Δp52/p36* sporozoites. The numbers of UIS4-positive (red histograms) and UIS4-negative (black histograms) EEFs were determined by fluorescence microscopy 24 hr post-infection.

The following figure supplement is available for figure 6:

**Figure supplement 1.** Genetic complementation with *p52* and *p36* from *P. falciparum* or *P. vivax* does not restore sporozoite infectivity in Pb*Δp52p36* parasites.

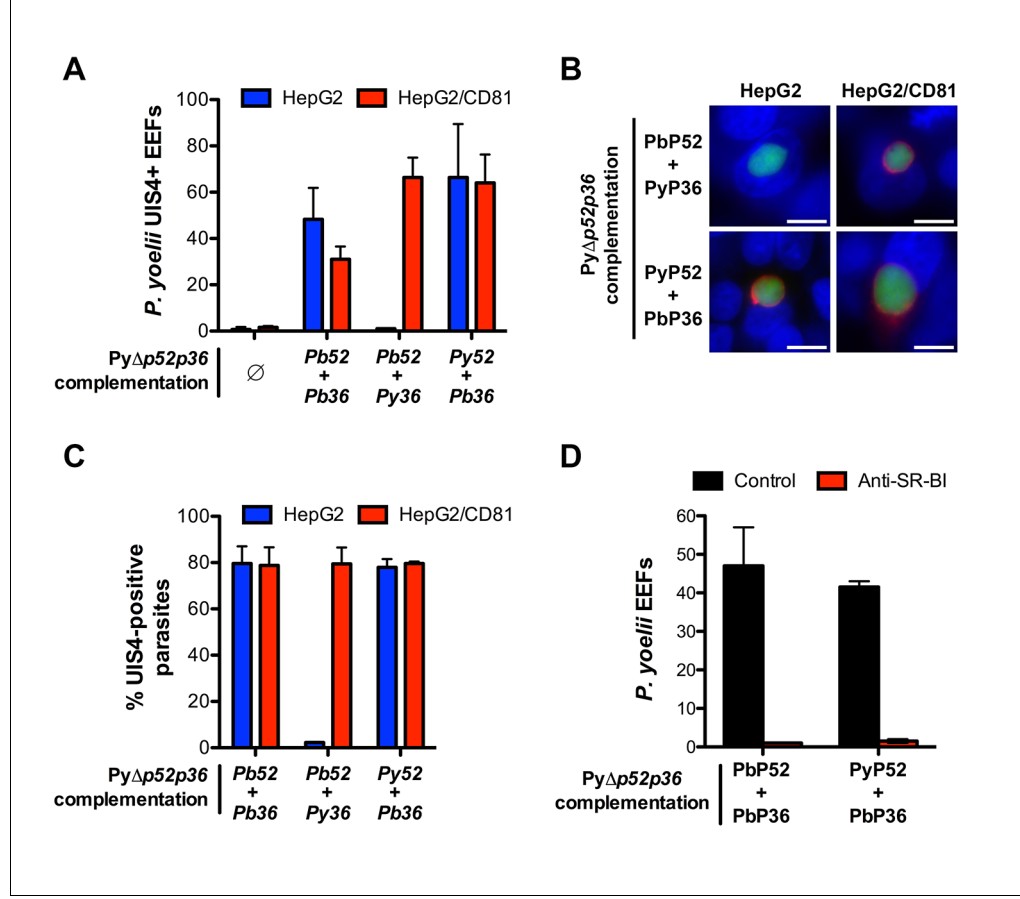

**Figure 7.** Transgenic *P. yoelii* sporozoites expressing *Pb*P36 can infect CD81-null cells via SR-BI. (**A**) HepG2 (blue) and HepG2/CD81 cells (red) were incubated with genetically complemented Py*Δp52/p36* sporozoites, and fixed 24 hr post-infection. The number of UIS4-positive vacuoles was then determined by immunofluorescence. (**B**) Immunofluorescence analysis of UIS4 expression in HepG2 or HepG2/CD81 cells infected with sporozoites of Py*Δp52/p36* parasites genetically complemented with P52 and P36 from *P. berghei* or *P. yoelii*. Cells were fixed with PFA, permeabilized, and stained with anti-UIS4 antibodies (red) and the nuclear stain Hoechst 33342 (blue). Parasites were detected based on GFP expression (green). Scale bars, 10 μm. (**C**) Quantification of UIS4 expression in HepG2 (blue) and HepG2/CD81 cells (red) infected with genetically complemented Py*Δp52/p36* sporozoites and processed as in B for immunofluorescence. (**D**) HepG2 cells were incubated with Py*Δp52/p36* sporozoites complemented with PbP36 and either PbP52 or PyP52, in the presence or absence of anti-SR-BI antibodies. Infected cultures were fixed 24 hr post-infection, and the number of EEFs was then determined by fluorescence microscopy.

contributes to *P. yoelii* infection through regulation of membrane cholesterol and CD81 expression, however our data in the HepG2/CD81 cell model with both *P. yoelii* and *P. berghei* clearly rule out a role of SR-BI during CD81-dependent sporozoite entry. For the first time, we also analysed the role of host factors during *P. vivax* sporozoite infection. We found that, in contrary to *P. falciparum*, antibodies against SR-BI but not against CD81 inhibit infection of primary human hepatocyte cultures by *P. vivax* sporozoites, illustrating that the two main species causing malaria in humans use distinct routes to infect hepatocytes.

SR-BI and CD81 both have been shown to bind the HCV envelope glycoprotein E2 (*Pileri et al., 1998*; *Scarselli et al., 2002*; *Bartosch et al., 2003*), and act in a sequential and cooperative manner to mediate virus entry (*Kapadia et al., 2007*), together with several additional entry factors (*Colpitts and Baumert, 2016*; *Evans et al., 2007*; *Ploss et al., 2009*). By contrast, as shown here, SR-BI and CD81 operate independently during *Plasmodium* sporozoite infection. Remarkably, *P. berghei* can use alternatively CD81 or SR-BI to infect cells, which is surprising given that CD81 and SR-

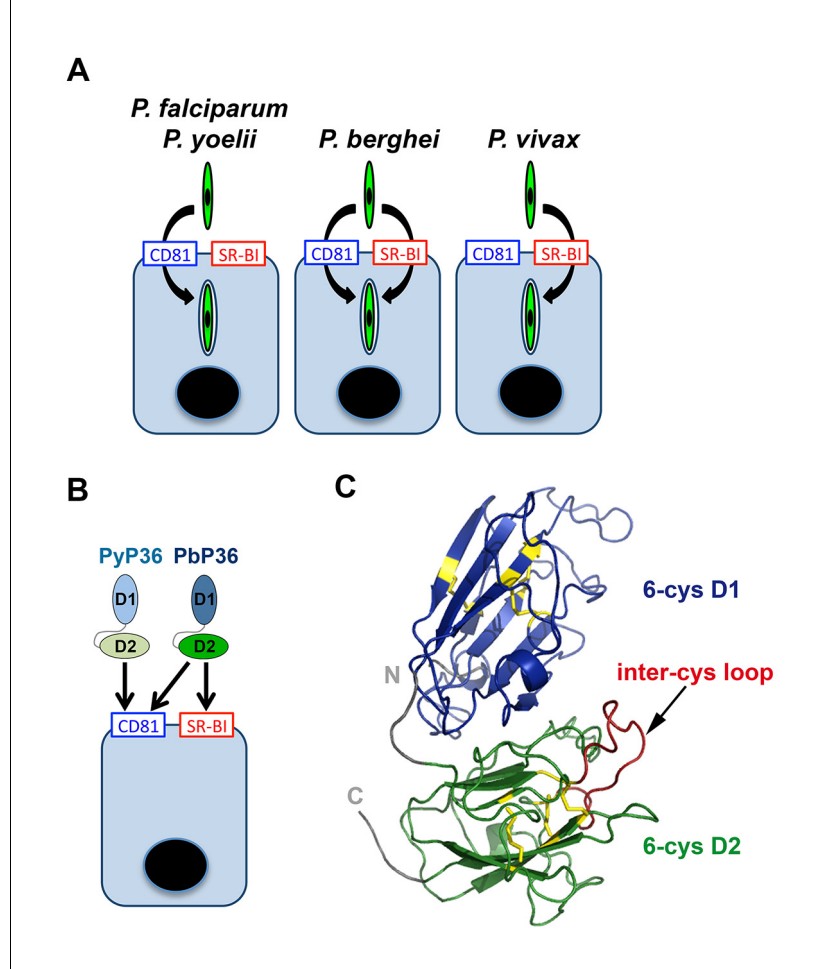

**Figure 8.** Model of host cell entry pathways for *Plasmodium* sporozoites. (**A**) Host cell membrane proteins CD81 and SR-BI define two independent entry routes for *Plasmodium* sporozoites. *P. falciparum* and *P. yoelii* sporozoites require CD81 for infection, whereas *P. vivax* sporozoites infect hepatocytes using SR-BI. *P. berghei* sporozoites can enter cells alternatively via CD81 or SR-BI. (**B**) The 6-cysteine domain protein P36 determines host cell receptor usage during *P. yoelii* and *P. berghei* sporozoite invasion. Whilst PyP36 supports only CD81-dependent sporozoite entry, PbP36 mediates sporozoite invasion through both CD81- and SR-BI-dependent pathways. (**C**) Model of the 3D structure of *P. berghei* P36, established based on the crystal structure of PfP12 (2YMO). In the ribbon diagram, the tandem 6-cysteine domains are shown in blue (D1) and green (D2), respectively, and the cysteine residues and disulphide bonds in yellow. The loop located between the third and fourth cysteine residues of the D2 domain (inter-cys loop) is indicated in red.

The following figure supplement is available for figure 8:

**Figure supplement 1.** P36 protein sequence analysis.

BI are structurally unrelated. CD81 belongs to the family of tetraspanins, which are notably characterized by their propensity to dynamically interact with other membrane proteins and organize tetraspanin-enriched membrane microdomains (*Charrin et al., 2014*). CD81 might play an indirect role during *Plasmodium* sporozoite entry, possibly by interacting with other host receptors within these microdomains (*Silvie et al., 2006b*; *Charrin et al., 2009a*, *2009b*). Interestingly, SR-BI is structurally related to CD36 (*Neculai et al., 2013*), which is known to bind several *Plasmodium* proteins, including *P. falciparum* erythrocyte membrane protein 1 (PfEMP1) in the context of cytoadherence of infected erythrocytes to endothelial cells (*Baruch et al., 1996*; *Oquendo et al., 1989*; *Barnwell et al., 1989*). CD36 was also reported to interact with *P. falciparum* sequestrin (also called

LISP2) (*Ockenhouse et al., 1991*), a member of the 6-cys protein family involved in parasite liver stage development (*Orito et al., 2013*; *Annoura et al., 2014*), and is a major determinant of *P. berghei* asexual blood stage sequestration (*Franke-Fayard et al., 2005*). CD36 was shown to be dispensable for mouse hepatocyte infection by *P. yoelii* and *P. berghei* sporozoites (*Sinnis and Febbraio, 2002*). However, it is conceivable that SR-BI, which shares a similar 3D structure as CD36 (*Neculai et al., 2013*), may interact with parasite proteins expressed by sporozoites, such as the 6-cys protein P36.

The 6-cys protein family is characterized by the presence of a cysteine-rich domain, the 6-cysteine (6-cys) or s48/45 domain. *Plasmodium* spp. possess a dozen 6-cys proteins, which perform important functions in different life cycle stages, and are often located on the parasite surface, consistent with a role in cellular interactions (*Annoura et al., 2014*). Previous studies have shown that *Plasmodium* P52 and P36 are crucial for infection of the liver by sporozoites (*van Dijk et al., 2005*; *Ishino et al., 2005*; *van Schaijk et al., 2008*), although it remained unclear whether their role was at the sporozoite entry step (*Ishino et al., 2005*; *Labaied et al., 2007*) or for maintenance of the PV post-entry (*van Dijk et al., 2005*; *VanBuskirk et al., 2009*; *Mikolajczak et al., 2014*). It should be noted that standard invasion assays, as performed in these studies, do not distinguish between sporozoite productive entry and non-productive invasion events associated with cell traversal, complicating the interpretation of phenotypic analysis of the mutants. Here, using GFP-expressing *p52* and *p36* mutants and a robust FACS-based invasion assay (*Risco-Castillo et al., 2015*), we unequivocally establish that *P. yoelii* and *P. berghei* sporozoites lacking P52 and P36 efficiently migrate through cells but do not commit to productive invasion, reproducing the phenotype observed upon blockage of CD81 or SR-BI.

Using a trans-species genetic complementation strategy, we identified P36 as a crucial parasite determinant of host receptor usage. Our data, combined with previous studies (*van Dijk et al., 2005*; *Ishino et al., 2005*; *van Schaijk et al., 2008*; *Kaushansky et al., 2015*), demonstrate that both P36 and P52 are necessary for sporozoite infection of hepatocytes, irrespective of the invasion route used by the parasite. Our study now reveals that P36 but not P52 is responsible for the phenotypic differences between *P. berghei* and *P. yoelii* sporozoites regarding host cell entry pathways. Importantly, *P. berghei* P36 mediates sporozoite entry via either CD81 or SR-BI, whereas *P. yoelii* P36 only supports CD81-dependent invasion (*Figure 8B*). These results strongly suggest that PbP36 contains specific structural determinants that confer the ability of the protein to interact with SR-BI or SR-BI-dependent molecules. *P. berghei* and *P. yoelii* P36 proteins share 88% identity and 97% similarity in their amino acid sequence (*Figure 8—figure supplement 1*). Structural modelling of PbP36, using the crystal structure of the 6-cys protein PfP12 (*Tonkin et al., 2013*) as a template, shows a typical beta sandwich fold for each of the tandem 6-cysteine domains (*Figure 8C*). Most of the divergent residues between PbP36 and PyP36 are located in the second 6-cys domain (*Figure 8—figure supplement 1*), including in a loop located between the third and fourth cysteine residues. In this particular loop, PfP36 and PvP36 contain an inserted sequence of 21 and 3 amino acids, respectively, which may affect their binding properties and functions.

Single gene deletions of *p52* or *p36* result in similar phenotypes as *p52/p36* double knockouts, suggesting that the two proteins act in concert (*van Dijk et al., 2005*; *Ishino et al., 2005*; *van Schaijk et al., 2008*; *Labaied et al., 2007*). In *P. falciparum* blood stages, the two 6-cys proteins P41 and P12 interact to form stable heterodimers on the surface of merozoites (*Tonkin et al., 2013*; *Taechalertpaisarn et al., 2012*; *Parker et al., 2015*). Whilst P12 is predicted to be GPI-anchored, P41 lacks a membrane-binding domain, similarly to P36. By analogy, we hypothesize that P36 may form heterodimers with other GPI-anchored 6-cys proteins, including P52. Our data show that complementation of PbΔ*p52/p36* parasites with P52 and P36 from *P. falciparum* or *P. vivax* does not restore sporozoite infectivity, supporting the idea that other yet unidentified parasite factors cooperate with P52 and P36 during invasion. In addition to P52 and P36, sporozoites express at least three other 6-cys proteins, B9, P12p and P38 (*Lindner et al., 2013*; *Swearingen et al., 2016*). Whereas gene deletion of *p38* causes no detectable phenotypic defect in *P. berghei* (*van Dijk et al., 2010*), B9 has been shown to play a critical role during liver stage infection, not only in *P. berghei* but also in *P. yoelii* and *P. falciparum* (*Annoura et al., 2014*). Whether B9, P38 and P12p associate with P52 and/or P36 and contribute to sporozoite invasion still deserves further investigations.

Several 6-cys proteins have been implicated in molecular interactions with host factors. As mentioned above, sequestrin was reported to interact with CD36 (*Ockenhouse et al., 1991*), although

the functional relevance of this interaction remains to be determined, as sequestrin is only expressed towards the end of liver stage development (*Orito et al., 2013*). Recent studies have shown that *P. falciparum* merozoites evade destruction by the human complement through binding of host factor H to the 6-cys protein Pf92 (*Rosa et al., 2016*; *Kennedy et al., 2016*). Pfs47 expressed by *P. falciparum* ookinetes plays a critical role in immune evasion in the mosquito midgut, by suppressing nitration responses that activate the complement-like system (*Molina-Cruz et al., 2013*; *Ramphul et al., 2015*). Pfs47 was proposed to act as a 'key' that allows the parasite to switch off the mosquito immune system by interacting with yet unidentified mosquito receptors ('lock') (*Molina-Cruz et al., 2015*). By analogy, based on our results, one could envisage P36 as a crucial determinant of a sporozoite 'key' that opens SR-BI and/or CD81-dependent 'locks' for entry into hepatocytes.

The function of P36 interaction with host cell receptors remains to be defined. P36 binding to SR-BI and/or CD81, either directly or indirectly, may participate in a signalling cascade that triggers rhoptry secretion and assembly of the moving junction, key events committing the parasite to host cell entry (*Besteiro et al., 2011*). Alternatively, P36 may induce signalling in the host cell by acting on SR-BI or other hepatocyte receptors. In this regard, Kaushansky *et al.* recently reported that sporozoites preferentially invade host cells expressing higher levels of the EphA2 receptor (*Kaushansky et al., 2015*). Interestingly, this preference was still observed with *p52*/*p36*-deficient parasites, strongly suggesting that there is no direct link between EphA2 and P52/P36-dependent productive invasion. However, the same study showed that P36 interferes with Ephrin A1-mediated EphA2 phosphorylation (*Kaushansky et al., 2015*), raising the possibility that P36 may affect EphA2 signalling indirectly, for example via SR-BI and the SRC pathway (*Naudin et al., 2014*; *Mineo and Shaul, 2003*).

In conclusion, our study reveals that host CD81 and SR-BI define two alternative pathways in human cells for sporozoite entry. Most importantly, we identified the parasite 6-cysteine domain protein P36 as a key determinant of host receptor usage during infection. These results pave the way toward the elucidation of the mechanisms of sporozoite invasion. The identification of the parasite ligands that mediate host cell entry may provide potential targets for the development of next-generation malaria vaccines. P36 is required for both CD81- and SR-BI-dependent sporozoite entry, suggesting that it may represent a relevant target in both *P. falciparum* and *P. vivax*. The understanding of host-parasite interactions may also contribute to novel therapeutic approaches. SR-BI-targeting agents have entered clinical development for prevention of HCV graft infection (*Felmlee et al., 2016*). Our data suggest that SR-BI-targeting strategies may be effective to prevent establishment of the liver stages of *P. vivax*, including the dormant hypnozoite forms.

## Materials and methods

### Experimental animals, parasites and cells

We used wild type *P. berghei* (ANKA strain, clone 15cy1) and *P. yoelii* (17XNL strain, clone 1.1), and GFP-expressing PyGFP and PbGFP parasite lines, obtained after integration of a GFP expression cassette at the dispensable *p230p* locus (*Manzoni et al., 2014*). *P. berghei* and *P. yoelii* blood stage parasites were propagated in female Swiss mice (6–8 weeks old, from Janvier Labs). *Anopheles stephensi* mosquitoes were fed on *P. berghei* or *P. yoelii*-infected mice using standard methods (*Ramakrishnan et al., 2013*), and kept at 21°C and 24°C, respectively. *P. berghei* and *P. yoelii* sporozoites were collected from the salivary glands of infected mosquitoes 21–28 or 14–18 days post-feeding, respectively. *A. stephensi* mosquitoes infected with *P. falciparum* sporozoites (NF54 strain) were obtained from the Department of Medical Microbiology, University Medical Centre, St Radboud, Nijmegen, the Netherlands. *P. vivax* sporozoites were isolated from *A. cracens* mosquitoes, 15–21 days after feeding on blood from infected patients on the Thailand-Myanmar border, as described (*Andolina et al., 2015*). HepG2 (ATCC HB-8065), HepG2/CD81 (*Silvie et al., 2006a*) and Hepa1-6 cells (ATCC CRL-1830) were checked for the absence of mycoplasma contamination and cultured at 37°C under 5% $CO_2$ in DMEM supplemented with 10% fetal calf serum and antibiotics (Life Technologies), as described (*Silvie et al., 2007*). HepG2 and HepG2/CD81 were cultured in culture dishes coated with rat tail collagen I (Becton-Dickinson, Le Pont de Claix, France). Primary human hepatocytes were isolated and cultured as described previously (*Silvie et al., 2004*).

## In vitro infection assays

Primary human hepatocytes ($5 \times 10^4$ per well in collagen-coated 96-well plates) were infected with *P. vivax* or *P. falciparum* sporozoites ($3 \times 10^4$ per well), as described (*Silvie et al., 2004*), and cultured for 5 days before fixation with cold methanol and immunolabeling of EEFs with antibodies specific for *Plasmodium* HSP70 (*Tsuji et al., 1994*). Nuclei were stained with Hoechst 33342 (Life Technologies). Host cell invasion by GFP-expressing *P. berghei* and *P. yoelii* sporozoites was monitored by flow cytometry (*Prudêncio et al., 2008*). Briefly, hepatoma cells ($3 \times 10^4$ per well in collagen-coated 96-well plates) were incubated with sporozoites ($5 \times 10^3$ to $1 \times 10^4$ per well). At different time points, cell cultures were washed, trypsinized and analyzed on a Guava EasyCyte 6/2L bench cytometer equipped with a 488 nm laser (Millipore), for detection of GFP-positive cells. To assess liver stage development, HepG2 or HepG2/CD81 cells were infected with GFP-expressing sporozoites and cultured for 24–36 hr before analysis either by FACS or by fluorescence microscopy, after fixation with 4% PFA and staining with antibodies specific for UIS4 (Sicgen) and Hoechst 33342. For antibody-mediated inhibition assays, we used polyclonal antisera against human SR-BI raised after genetic immunization of rabbits and rats (*Maillard et al., 2006*; *Zeisel et al., 2007*), and monoclonal antibodies against human SR-BI (NK-8H5-E3) (*Zahid et al., 2013*), human CD81 (1D6, Abcam) or mouse CD81 (MT81)(*Silvie et al., 2006b*).

## Small interfering RNA and plasmid transfection

We used small double stranded RNA oligonucleotides targeting human CD81 (5'-GCACCAAGTGCA TCAAGTA-3'), human SR-BI (5'-GGACAAGTTCGGATTATTT-3') or mouse CD81 (5'-CGTGTCACC TTCAACTGTA-3'). An irrelevant siRNA oligonucleotide targeting human CD53 (5'-CAACTTCGGAG TGCTCTTC-3') was used as a control. Transfection of siRNA oligonucleotides was performed by electroporation, as described (*Silvie et al., 2006a*). Following siRNA transfection, cells were cultured for 48 hr before flow cytometry analysis or sporozoite infection. Transfection of pcDNA3 plasmids encoding human CD81 (*Yalaoui et al., 2008b*) or SR-BI (*Maillard et al., 2006*) was performed 24 hr after siRNA using the Lipofectamine 2000 reagent (Invitrogen) according to the manufacturer's specifications. Following plasmid transfection, cells were cultured for an additional 24 hr before sporozoite infection.

## Constructs for targeted gene deletion of *p52* and *p36*

Pb*Δp52p36* and Pb*Δp36* mutant parasites were generated using a 'Gene Out Marker Out' strategy (*Manzoni et al., 2014*). For generation of Pb*Δp52p36* parasites, a 5' fragment of PbP52 gene (PBANKA_1002200) and a 3' fragment of PbP36 gene (PBANKA_1002100) were amplified by PCR from *P. berghei* ANKA WT genomic DNA, using primers PbP52rep1for (5'-TCCCCGCGGAATCG TGATGCTATGGATAACGTAACAC-3'), PbP52rep2rev (5'-ATAAGAATGCGGCCGCAAAAAGAGA-CAAACACACTTTGTGAACACC-3'), PbP36rep3for (5'-CCGCTCGAGTTAATATGTGATGTGTGT TAGAAGAGTGAGG-3') and PbP36rep4rev (5'-GGGGTACCTTGATATACATGCAACTTTTCACA TAGG-3'), and inserted into *SacII/NotI* and *XhoI/KpnI* restriction sites, respectively, of the GOMO-GFP vector. For generation of Pb*Δp36* parasites, a 5' and a 3' fragment of PbP36 gene were amplified by PCR from *P. berghei* ANKA WT genomic DNA, using primers PbP36repAfor (5'-AGC TGGAGCTCCACCGCGGGAAAAAAGGTTAACACATATATTGAAAAGC-3'), PbP36rep-Arev (5'-CGGCTGAGGGCGGCCGCAATCAAAAAAAATAATAAAAACAAATATATGTACACTCG-3'), and PbP36repBfor (5'-ATTAATTTCACTCGAGTATGTGATGTGTGTAGAAGAGTGAGG-3') and PbP36rep-Brev (5'-TATAGGGCGAATTGGGTACCGCACGCCGGAAAAATTACAATACAAATGG-3'), and inserted into *SacII/NotI* and *XhoI/KpnI* restriction sites, respectively, of the GOMO-GFP vector using the In-Fusion HD Cloning Kit (Clontech).

For generation of Py*Δp52p36* parasites, 5' and 3' fragments of PyP52 gene (PY17X_1003600) and a 3' fragment of PyP36 gene (PY17X_1003500) were amplified by PCR from *P. yoelii* 17XNL WT genomic DNA, using primers PyP52rep1for (5'-TCCCCGCGGAATCGCCATGCTATGGATAGTG TAGC-3'), PyP52rep2rev (5'-ATAAGAATGCGGCCGCCATTGAAGGGGGGAACAAATCGACG-3'), PyP52rep3for (5'-CCGCTCGAGTCAATATATGCCCACTATTCGAATTTTTGG-3'), PyP52rep4rev (5'-GGGGTACCTTATTGATATGCATGCAACTTTCACATAGG-3'), PyP36repFor (5'-ATAAGAA TGCGGCCGCAAAATGCAAGGCGCCCGTTTAGAACC-3') and PyP36repRev (5'-CCGGAATTCA-CAAAAAGATGCTACTGTGAAAAGCTCACC-3'). The fragments were inserted into *SacII/NotI*, *XhoI/*

*Kpn*I and *Not*I/*Eco*RI restriction sites, respectively, of a GOMO vector backbone containing mCherry and hDHFR-yFCU cassettes. The resulting targeting constructs were verified by DNA sequencing (GATC Biotech), and were linearized with *Sac*II and *Kpn*I before transfection.

Wild type *P. berghei* ANKA blood stage parasites were transfected with *pbp52p36* and *pbp36* targeting constructs using standard transfection methods (Janse et al., 2006). GFP-expressing parasite mutants were isolated by flow cytometry after positive and negative selection rounds, as described (Manzoni et al., 2014). PyGFP blood stage parasites were transfected with a *pyp52pyp36* targeting construct and a GFP-expressing drug selectable marker-free PyΔp52p36 mutant line was obtained using a two steps 'Gene Out Marker Out' strategy. Correct construct integration was confirmed by analytical PCR using specific primer combinations.

## Constructs for genetic complementation of Δ*p52p36* mutants

For genetic complementation experiments, we used centromeric plasmids to achieve stable transgene expression from episomes (Iwanaga et al., 2010). Complementation plasmids were obtained by replacing the GFP cassette of pCEN-SPECT2 plasmid (kindly provided by Dr S. Iwanaga) with a P52/P36 double expression cassette. Four complementing plasmids were generated, allowing expression of PbP52/PbP36, PyP52/PyP36, PbP52/PyP36 or PyP52/PbP36.

The centromeric plasmid constructs were assembled by In-Fusion cloning of 4 fragments in two-steps, into KpnI/SalI restriction sites of the pCEN-SPECT2 plasmid. For this purpose, fragments corresponding to the promoter region of PbP52 (insert A, 1.5 kb), the open reading frame (ORF) of PbP52 (insert B, 1.7 kb), the 3' untranslated region (UTR) of PbP52 and promoter region of PbP36 (insert C, 1.5 kb), the ORF and 3' UTR of PbP36 (insert D, 2 kb), the promoter region of PyP52 (insert E, 1.5 kb), the open reading frame (ORF) of PyP52 (insert F, 1.7 kb), the 3' untranslated region (UTR) of PyP52 and promoter region of PyP36 (insert G, 1.6 kb), and the ORF and 3' UTR of PyP36 (insert H, 2 kb) were first amplified by PCR from *P. berghei* or *P. yoelii* WT genomic DNA, using the following primers: Afor (5'-TATAGGGCGAATTGGGTACCTTCACATGCATAAACCCGAAGTGTGC-3'), Arev (5'-GAAAAAAGCAGCTAGCTTGCTTTAATGTAGAAAAAATATTTATGGATTTGG-3'), Bfor (5'-ATTAAAGCAAGCTAGCAATATTACATTTGTGGTAAGGTAAAAC-3'), Brev (5'-GAAGAGGTAC-CAAAAAGGTTTTGCCAAAATG-3'), Cfor (5'-TTTTGGTACCTCTTCTTCTTATTATGAGG-3'), Crev (5'-GAAAAAAGCAGCTAGCAGAAAGAAACAACAGTTATCGTAATAAAG-3'), Dfor (5'-GCTAGCTGCTTTTTTCTTGAATCGACAATTATAATACTGAGGC-3'), Drev (5'-TACAAGCATCGTCGACATTGCCATTACAATATGCTATAATCTG-3'), Efor (5'-TATAGGGCGAATTGGGTACCTGCACATGCATAAACTCGAAGTGTGC-3'), Erev (5'-AAAAAAGCAGCTAGCTTGCTTTAATGTAGAAAAAATATTTATGTATTTGG-3'), Ffor (5'-ATTAAAGCAAGCTAGAATATTGCATTTGTGGTAAGGCAAATC-3'), Frev (5'-GAAGACGTACCAAACATATTTTGCCAAAATG-3'), Gfor (5'-GTTTGGTACGTCTTCTTCTTATTATGAGG-3'), Grev (5'-GAAAAAAGCAGCTAGGATAACTGTCGATTCAAAGAAACAACC-3'), Hfor (5'-GCTAGCTGCTTTTTTTATACTTGAAGCATTTTTGTTGACTCTACC-3'), Hrev (5'-TACAAGCATCGTCGACATTACCATTACGATATGCTATAATCTG-3'). Cloning of fragments A and D followed by B and C into the pCEN vector resulted in the PbP52/PbP36 complementation plasmid. Cloning of fragments E and H followed by F and G into the pCEN vector resulted in the PyP52/PyP36 complementation plasmid. Cloning of fragments A and D followed by F and G into the pCEN vector resulted in the PyP52/PbP36 complementation plasmid. Cloning of fragments E and H followed by B and C into the pCEN vector resulted in the PbP52/PyP36 complementation plasmid. The centromeric plasmids for expression of *P. falciparum* and *P. vivax* P52 and P36 were assembled by In-Fusion cloning of 5 fragments in two steps. For this purpose, fragments corresponding to the promoter region of *PbP52* (insert B1, 1.9 kb), the 3' UTR of *PbP52* and promoter region of *PbP36* (insert B2, 1.6 kb), the 3' UTR of *PbP36* (insert B3, 2 kb), the ORF of *PfP52* (insert F1, 1.4 kb), the ORF of *PfP36* (insert F2, 1.1 kb), the ORF of *PvP52* (insert V1, 1.4 kb) and the ORF of *PvP36* (insert V2, 1.1 kb), were first amplified by PCR from *P. berghei*, *P. falciparum* or *P. vivax* genomic DNA, using the following primers: B1for (5'-tatagggcgaattgggtaccTTCACATGCATAAACCCGAAGTGTGC-3'), B1rev (5'-gctagcTTACTATTATTCTCAAAATGTGTATCACATTG-3'), B2for (5'-ATCACAATATGTGCATAGTGTCAATATGCC-3'), B2rev (5'-AATCAAAAAAAATAATAAAAACAAATATATGTACACTCG-3'), B3for (5'-TAATAGTAAgctagcTATGTGATGTGTGTAGAAGAGTGAGGGAG-3'), B3rev (5'-tacaag-catcgtcgacATTGCCATTACAATATGCTATAATCTG-3'), F1for (5'-ATAATAGTAAgctagcAAAATGTATGTATTGGTGCTTATTCATATGTG-3'), F1rev (5'-TGCACATATTGTGATTTATGTTGAATATATATATATTAAAAATATGAATAATATTAAG-3'), F2for (5'-TTATTTTTTTTGATTATGGCTTATAATA

TTTGGGAGGAATATATAATGG-3'), F2rev (5'-ACATCACATAgctagcCTAACTTTCTACAGTTTTATTTA TGTTAAATAAACC-3'), V1for (5'-ATAATAGTAAgctagcAAAATGAGGCGGATTCTGCTGGGCTGC TTGG-3'), V1rev (5'-TGCACATATTGTGATTTACAGGGACGAGAAACCCGCGTAG-3'), V2for (5'-TTA TTTTTTTTTGATTATGAGCACATGCCTTCCAGTAGTGTGG-3'), and V2rev (5'-ACATCACATAgctagcT-CACACCGCTTCAACCGCTGCG-3'). An intermediate vector was first assembled by cloning inserts B1 and B3 into *Kpn*I/*Sal*I restriction sites of the pCEN-SPECT2 plasmid. Subsequently, In-Fusion cloning of inserts F1, B2 and F2 or inserts V1, B2 and V2 into the *Nhe*I restriction site of the intermediate vector resulted into PfP52/PfP36 and PvP52/PvP36 expression centromeric plasmids, respectively. All constructs were verified by DNA sequencing (GATC Biotech) before transfection.

## Parasite transfection and selection

For double crossover replacement of P52 and P36 genes and generation of the Pb*Δp52p36*, Pb*Δp36* and Py*Δp52p36* parasite lines, purified schizonts of wild type *P. berghei* ANKA or PyGFP were transfected with 5–10 µg of linearized construct by electroporation using the AMAXA Nucleo-fectorTM device (program U33), as described elsewhere (*Janse et al., 2006*), and immediately injected intravenously in the tail of one mouse. The day after transfection, pyrimethamine (70 or 7 mg/L for *P. berghei* and *P. yoelii*, respectively) was administered in the mouse drinking water, for selection of transgenic parasites. Pure transgenic parasite populations were isolated by flow cytometry-assisted sorting of GFP and mCherry-expressing blood stage parasites on a FACSAria II (Becton-Dickinson), transferred into naïve mice, treated with 1 mg/ml 5-fluorocytosine (Meda Pharma) in the drinking water, and sorted again for selection of GFP+ parasites only, as described (*Manzoni et al., 2014*). In the case of the Py*Δp52p36* mutant, GFP+ mCherry+ recombinant parasites were first cloned by injection of limiting dilutions into mice prior to the negative selection step. For genetic complementation of the mutants, purified schizonts of Pb*Δp52p36* and Py*Δp52p36* parasites were transfected with 5 µg of centromeric plasmids, followed by positive selection of transgenic parasites with pyrimethamine, as described above.

## Parasite genotyping

Parasite genomic DNA was extracted using the Purelink Genomic DNA Kit (Invitrogen), and analysed by PCR using primer combinations specific for WT and recombined loci. For genotyping of Pb*Δp52p36* parasites, we used primer combinations specific for the WT Pb*p52* locus (5'-AATGAGA TGTCAAAAAATATAGTGCTTCC-3' and 5'-AAATGAGCAGTTTCTTCTACGTTGTTTCC-3'), for the 5' region of the recombined locus (5'-TATGTTTGGAATATCAGGACAAGGCATGG-3' and 5'-TAATAA TTGAGTCTTTAGTAACGAATTGCC-3'), and for the 3' region of the recombined locus before (5'-A TCGTGGAACAGTACGAACGCGCCGAGG-3' and 5'-ATTGGACGTTTATTATTATTGCAAAAGCG-3') or after excision of the selectable marker (5'-GATGGAAGCGTTCAACTAGCAGACC-3' and 5'-A TTGGACGTTTATTATTATTGCAAAAGCG-3'). For genotyping of Pb*Δp36* parasites, we used primer combinations specific for the WT Pb*p36* locus (5'-GAGTTCGCACGCCATATTAACACG-3' and 5'-CCATGATGAGATGCTAAATCGGG-3'), for the 5' region of the recombined locus (5'-GGAAGCA TCATACAAAAAAGAAAGC-3' and 5'-TAATAATTGAGTCTTTAGTAACGAATTGCC-3'), and for the 3' region of the recombined locus before (5'-ATCGTGGAACAGTACGAACGCGCCGAGG-3' and 5'-CGTTATCTCTTTTTTTACTCATTAAGTATTG-3') or after excision of the selectable marker (5'-GA TGGAAGCGTTCAACTAGCAGACC-3' and 5'-CGTTATCTCTTTTTTTACTCATTAAGTATTG-3'). For genotyping of Py*Δp52p36* parasites, we used primer combinations specific for the WT PyP52 locus (5'-ACTATATTTCAATTGGAGACATGTGG-3' and 5'-ATGCAAAAAAAAGTTATCATTGCTAGTTGG-3'), for the 5' region of the recombined locus (5'-GTATGTTTGGAATGCCAGGATATGACATGG-3' and 5'-CCGGAATTCACAAAAAGATGCTACTGTGAAAAGCTCACC-3'), and for the 3' region of the recombined locus before (5'-AGTTACACGTATATTACGCATACAACGATG-3' and 5'-TAAGCATATA TTGTATATTTGCCTTGTCC-3') or after excision of the selectable marker (5'-GTATGTTTGGAA TGCCAGGATATGACATGG-3' and 5'-AATCTGATATGATAAATTATGGTATTGGAC-3').

## Bioinformatic and structural analysis

Amino-acid sequences of the P36 proteins from *P. falciparum* (379 aa, gi:296004390, PF3D7_0404400), *P. vivax* (320 aa, gi:156094683, PVX_001025), *P. berghei* (352 aa, gi:991456178, PBANKA_1002100) and *P. yoelii* (356 aa, gi:675237743, PY17X_1003500) were obtained from

Genbank. Sequence alignments were carried out using Clustal Omega (http://www.ebi.ac.uk/Tools/msa/clustalo/). A 3D model of *P. berghei* P36 was generated by the prediction program I-Tasser (*Yang et al., 2015*), using the 3D structure of Pf12 (Pf12short in ref [*Tonkin et al., 2013*]), which contains two 6-cys domains D1 and D2 arranged in tandem, as a template (PDB access code 2YMO). The 3D model for PbP36 was then superimposed to the template Pf12short and visually inspected using the program Coot (*Emsley and Cowtan, 2004*), and the rotamers for the Cys residues adjusted such that the three disulfides bonds for each domain were formed following the pattern C1-C2, C3-C6 and C4-C5.

## Statistical analysis

Statistical significance was assessed by non-parametric analysis using the Mann-Whitney U and Kruskal-Wallis tests. All statistical tests were computed with GraphPad Prism 5 (GraphPad Software). Significance was defined as $p < 0.05$ (ns, statistically non-significant; *$p < 0.05$; **$p < 0.01$). In vitro experiments were performed at least three times, with a minimum of three technical replicates per experiment.

## Ethics statement

All animal work was conducted in strict accordance with the Directive 2010/63/EU of the European Parliament and Council 'On the protection of animals used for scientific purposes'. The protocol was approved by the Charles Darwin Ethics Committee of the University Pierre et Marie Curie, Paris, France (permit number Ce5/2012/001). Blood samples were obtained from *P. vivax*-infected individuals attending the Shoklo Malaria Research Unit (SMRU) clinics on the western Thailand-Myanmar border, after signature of a consent form (*Andolina et al., 2015*). Primary human hepatocytes were isolated from healthy parts of human liver fragments, which were collected from adult patients undergoing partial hepatectomy (Service de Chirurgie Digestive, Hépato-Bilio-Pancréatique et Transplantation Hépatique, Groupe Hospitalier Pitié-Salpêtrière, Paris, France). The collection and use of this material were undertaken in accordance with French national ethical guidelines under Article L. 1121–1 of the 'Code de la Santé Publique', and approved by the Institutional Review Board (Comité de Protection des Personnes) of the Centre Hospitalo-Universitaire Pitié-Salpêtrière, Assistance Publique-Hôpitaux de Paris, France.

## Acknowledgements

We thank Maurel Tefit, Thierry Houpert, Prapan Kittiphanakun and Saw Nay Hsel for rearing of mosquitoes in Paris and at SMRU; Geert-Jan van Gemert and Robert W Sauerwein for providing *P. falciparum*-infected mosquitoes; Bénédicte Hoareau-Coudert (Flow Cytometry Core CyPS) for parasite sorting by flow cytometry; Valérie Soulard, Audrey Lorthiois and Morgane Mitermite for technical assistance; Maryse Lebrun for critically reading the manuscript; and Shiroh Iwanaga for kindly providing the pCEN-SPECT2 plasmid. This work was funded by the European Union (FP7 Marie Curie grant PCIG10-GA-2011–304081, FP7 PathCo Collaborative Project HEALTH-F3-2012-305578, ERC-AdG-2014–671231-HEPCIR, FP7 HepaMab, and Interreg IV-Rhin Supérieur-FEDER Hepato-Regio-Net 2012, H2020-2015-667273-HEP-CAR), the Laboratoires d'Excellence ParaFrap and HepSYS (ANR-11-LABX-0024 and ANR-10-LABX-0028), and the National Centre for the Replacement, Refinement and Reduction of Animals in Research (NC3Rs, project grant NC/L000601/1). SMRU is part of the Mahidol-Oxford University Research Unit supported by the Wellcome Trust of Great Britain. GM and MG were supported by a 'DIM Malinf' doctoral fellowship awarded by the Conseil Régional d'Ile-de-France.

## Additional information

### Funding

| Funder | Author |
| --- | --- |
| European Commission | Dominique Mazier |
| | Thomas F Baumert |
| | Olivier Silvie |

| Agence Nationale de la Recherche | Mirjam B Zeisel<br>Dominique Mazier<br>Thomas F Baumert<br>Olivier Silvie |
| --- | --- |
| National Centre for the Replacement, Refinement and Reduction of Animals in Research | Olivier Silvie |
| Conseil Régional, Île-de-France | Giulia Manzoni<br>Marion Gransagne |
| Wellcome | François Nosten |

The funders had no role in study design, data collection and interpretation, or the decision to submit the work for publication.

## Author contributions

GM, CM, Formal analysis, Validation, Investigation, Visualization, Methodology, Writing—review and editing; ST, SB, MGra, MT, MGra, Investigation, Visualization, Methodology, Writing—review and editing; JL, Formal analysis, Visualization, Writing—review and editing; CA, Resources, Investigation, Methodology, Writing—review and editing; J-FF, Investigation, Methodology, Writing—review and editing; MBZ, TH, ER, GS, FN, TFB, Resources, Formal analysis, Writing—review and editing; DM, Formal analysis, Writing—review and editing; OS, Conceptualization, Formal analysis, Supervision, Funding acquisition, Validation, Investigation, Visualization, Methodology, Writing—original draft, Project administration, Writing—review and editing

## Author ORCIDs

Thierry Huby, http://orcid.org/0000-0001-6634-551X
Eric Rubinstein, http://orcid.org/0000-0001-7623-9665
François Nosten, http://orcid.org/0000-0002-7951-0745
Olivier Silvie, http://orcid.org/0000-0002-0525-6940

## Ethics

Human subjects: Blood samples were obtained from *P. vivax*-infected individuals attending the Shoklo Malaria Research Unit (SMRU) clinics on the western Thailand-Myanmar border, after signature of a consent form. Primary human hepatocytes were isolated from healthy parts of human liver fragments, which were collected from adult patients undergoing partial hepatectomy (Service de Chirurgie Digestive, Hépato-Bilio-Pancréatique et Transplantation Hépatique, Groupe Hospitalier Pitié-Salpêtriere, Paris, France). The collection and use of this material were undertaken in accordance with French national ethical guidelines under Article L. 1121-1 of the 'Code de la Santé Publique', and approved by the Institutional Review Board (Comité de Protection des Personnes) of the Centre Hospitalo-Universitaire Pitié-Salpêtrière, Assistance Publique-Hopitaux de Paris, France.

Animal experimentation: All animal work was conducted in strict accordance with the Directive 2010/63/EU of the European Parliament and Council 'On the protection of animals used for scientific purposes'. The protocol was approved by the Charles Darwin Ethics Committee of the University Pierre et Marie Curie, Paris, France (permit number Ce5/2012/001).

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
