## [Decision Letter]

Thank you for submitting your article "*Plasmodiu*m P36 determines host cell receptor usage during sporozoite invasion" for consideration by *eLife*. Your article has been reviewed by three peer reviewers, and the evaluation has been overseen by a Reviewing Editor and Wendy Garrett as the Senior Editor. The following individual involved in review of your submission has agreed to reveal their identity: Rita Tewari (Reviewer #2).

The reviewers have discussed the reviews with one another and the Reviewing Editor has drafted this decision to help you prepare a revised submission.

Summary:

There has been a lot of confusion in the field of *Plasmodium* sporozoite invasion into hepatocytes. Mainly two receptors (CD81 and SRBI) have been discussed to be involved but conflicting results have been published concerning their role. In series of very elegant experiments, the current study by Manzoni et al. provides a very robust explanation for the observed phenomenon. They show that CD81 and SR-BI do not sequentially promote *Plasmodium* infection as is the case for HCV. Instead, the authors demonstrate that closely related human and rodent *Plasmodium* species prefer different receptors for invasion in vitro. In addition, they show that a GPI-anchored sporozoite surface protein P36 is indeed the ligand that binds to either of both receptors depending on the parasite species. Importantly, of the four species investigated, only *P. berghei* sporozoites can bind both receptors.

This comprehensive study examines four *Plasmodium* species, including the important but difficult to study *P. vivax*, for which no data on invasion pathway utilisation have previously been published. Each point they make is well supported by a combination of experimental approaches.

Essential revisions:

To improve the quality of the manuscript, the reviewers felt that the authors should provide responses to clarify the questions listed below.

1) While the current manuscript appears to be consistent with an earlier and less complete analysis by Sylvie et al. (2007) introducing some of the reagents used in the current study, I am left wondering how their current data may be reconciled with the 2008 Cell Host Microbe paper by Yalaoui et al., who use primary hepatocytes from SR-BI KO and hypomorphic mice to implicate roles for SR-BI and CD81 for both Pb and Py. Can the authors comment how do the data produced here in cell lines relate to any data available from rodent primary cells and in vivo experiments using wt and KO mouse lines?

2) While the initial experiments of the manuscript have been performed with *P. vivax* and *P. falciparum* (Figure 1), these species have not been discussed in the rest of the manuscript. While it is clear that *P. vivax* is challenging, the reviewer felt that introducing *P. falciparum* into the model presented in Figure 8 would have strengthened the manuscript. As *P. falciparum* is routinely used in the PI's lab, are there particular reasons for excluding this species from further analyses?

3) Similarly, P36 and P52 mutant have been reported in *P. falciparum*, what was the reason of not exploiting them in this study? In addition, 6-cys domain family has two other members B9 and sequestrin that are also involved in liver stage development (Annoura et al. FASEB 2014). As the authors did not discuss them much it remains unclear how these molecules relate to this study. I am less convinced with P36 as the major determinant in the absence of these two other molecules been studied.

4) It has been shown by Mikolaijczak et al. in Molecular Therapy 2014 that double knockout P52/P36 parasites still invade the hepatocytes. How will the authors explain that P36 is the major determinant in the light of these even in *P. Falciparum*? Introducing at least some discussion on these results will strengthen the manuscript.

---

## [Author Response]

*Essential revisions:*

*To improve the quality of the manuscript, the reviewers felt that the authors should provide responses to clarify the questions listed below.*

*1) While the current manuscript appears to be consistent with an earlier and less complete analysis by Sylvie et al. (2007) introducing some of the reagents used in the current study, I am left wondering how their current data may be reconciled with the 2008 Cell Host Microbe paper by Yalaoui et al., who use primary hepatocytes from SR-BI KO and hypomorphic mice to implicate roles for SR-BI and CD81 for both Pb and Py. Can the authors comment how do the data produced here in cell lines relate to any data available from rodent primary cells and* in vivo *experiments using wt and KO mouse lines?*

We thank the reviewers for raising an important issue that we have now clarified in the revised manuscript. Our data show that antibody-mediated neutralization of SR-BI has no impact on *P. falciparum* and *P. yoelii* sporozoite infection, in primary human hepatocytes and HepG2/CD81 cells, respectively. These results are consistent with those obtained by Foquet et al. using a humanized mouse model (Foquet et al. J Antimicrob Chemother 2015, PMID 25656410), and also with the data from Yalaoui et al. showing that antibodies against SR-BI do not inhibit *P. yoelii* or *P. falciparum* sporozoite infection when administered together with sporozoites (Yalaoui et al. Cell Host & Microbe 2008, PMID 18779054). In the same study, the authors observed a reduction of infection upon pre-incubation of cells for several hours with anti-SR-BI antibodies, or in primary cultures of hepatocytes isolated from SR-BI-deficient mice. The study further showed that reduced infection was due to an indirect effect on CD81 expression in primary hepatocyte cultures, through interference with the membrane cholesterol levels. However, one should note that SR-BI knockout mice remain susceptible to *P. yoelii* sporozoite infection in vivo, in sharp contrast with the CD81-knockout mice that are totally refractory (Silvie et al. Nat Med 2003, PMID 12483205). Our data using RNA interference and three different neutralizing antibodies unequivocally demonstrate that SR-BI is dispensable during infection with *P. yoelii* and *P. falciparum,* in HepG2/CD81 cells and primary human hepatocytes, respectively, ruling out a role of SR-BI in CD81-dependent parasite entry.

This is an important point that is now explicitly discussed in the revised text: “Our results are also consistent with the study from Yalaoui et al. showing that in primary mouse hepatocytes antibodies against SR-BI do not inhibit *P. yoelii* infection when co-incubated together with sporozoites (Yalaoui et al., 2008). In the same study, the authors proposed a model where SR-BI indirectly contributes to *P. yoelii* infection through regulation of membrane cholesterol and CD81 expression. However, our data in the HepG2/CD81 cell model with both *P. yoelii* and *P. berghei* clearly rule out a role of SR-BI during CD81-dependent sporozoite entry”.

*2) While the initial experiments of the manuscript have been performed with P. vivax and P. falciparum (Figure 1), these species have not been discussed in the rest of the manuscript. While it is clear that P. vivax is challenging, the reviewer felt that introducing P. falciparum into the model presented in Figure 8 would have strengthened the manuscript. As P. falciparum is routinely used in the PI's lab, are there particular reasons for excluding this species from further analyses?*

Our initial experiments revealed the implication of SR-BI (but not CD81) during *P. vivax* sporozoite infection. To dissect in more details the role of SR-BI during infection, we turned to more tractable rodent models, where transgenic GFP-expressing parasite lines combined with highly susceptible cell lines (HepG2 and HepG2/CD81) provide robust experimental setups to explore the kinetics of infection, notably by flow cytometry. The rationale for using rodent parasites to dissect the contribution of SR-BI is now clearly indicated: “In order to investigate in more details the role of SR-BI during sporozoite entry, we used the more tractable rodent malaria parasite *P. berghei*.”

Nevertheless, we agree with the reviewers that addressing the role of 6-cystein proteins from *P. falciparum* and *P. vivax* is highly relevant with regards to the model that we propose in Figure 8. To address this question, we used our trans-species genetic complementation strategy and complemented PbΔ*p52/p36* parasites with P52 and P36 from either *P. falciparum* or *P. vivax*. Strikingly, both constructs did not restore infectivity of the transgenic mutant, irrespective of the entry route. In particular, complementation did not restore infectivity to primary human hepatocytes, the only cell culture system that sustains robust productive invasion of human malaria sporozoites. Therefore, it was not possible to conclude on the role of *P. falciparum* and *P. vivax* P52 and P36 in relation with the host cell entry pathways using this system. Nevertheless, the observation that P52/P36 from *P. falciparum* and *P. vivax* are not sufficient to complement PbΔ*p52/p36* parasites strongly suggest that additional factors are involved for sporozoite host cell invasion (see point 3 below). This new data set has been included in the revised manuscript as a new Figure 6—figure supplement 1. The manuscript text has been modified accordingly in the Results section: “Complementation of PbΔ*p52/p36* parasites with either *P. falciparum* or *P. vivax* P52 and P36 coding sequences did not restore infectivity of transgenic sporozoites, not only in HepG2 and HepG2/CD81 cells (Figure 6—figure supplement 1), but also in primary human hepatocytes, the most permissive cellular system for human malaria sporozoites in vitro (Figure 6—figure supplement 1). Hence it was not possible using this approach to assess the function of *P. falciparum* or *P. vivax* P52 and P36 in transgenic *P. berghei* sporozoites”, in the Discussion: “Our data show that complementation of PbΔ*p52/p36* parasites with P52 and P36 from *P. falciparum* or *P. vivax* does not restore sporozoite infectivity, supporting the idea that other yet unidentified parasite factors cooperate with P52 and P36 during invasion”, and in the Materials and methods (subsection “Constructs for genetic complementation of Δp52p36 mutants”).

*3) Similarly, P36 and P52 mutant have been reported in P. falciparum, what was the reason of not exploiting them in this study? In addition, 6-cys domain family has two other members B9 and sequestrin that are also involved in liver stage development (Annoura et al. FASEB 2014). As the authors did not discuss them much it remains unclear how these molecules relate to this study. I am less convinced with P36 as the major determinant in the absence of these two other molecules been studied.*

We agree with the reviewers that other factors, in addition to P36, likely participate in sporozoite entry. Our results showing that usage of host cell entry pathways (CD81 versus SR-BI) can be changed by swapping P36 sequence between *P. berghei* and *P. yoelii* clearly demonstrate that this protein is one key determinant of host receptor usage. On the other hand, P52 and P36 from *P. falciparum* or *P. vivax* fail to rescue infectivity of mutant *P. berghei* sporozoites (new Figure 6—figure supplement 1), strongly supporting the implication of additional factors. P36 harbors no membrane-binding domain, and may interact with GPI-anchored P52 or other sporozoite 6-cys proteins, similarly to P41 and P12, which form heterodimers on the surface of *P. falciparum* merozoites. Sporozoites express three other 6-cys proteins, B9, P12p and P38. B9 was reported to be essential for liver stage development but not invasion (Annoura et al. FASEB J 2014, PMID 24509910). Whether P38 and/or P12p participate in invasion and interact with P36 (and P52) deserves further investigations.

We now discuss these considerations in the revised text: “Single gene deletions of *p52* or *p36* result in similar phenotypes as *p52/p36* double knockouts, suggesting that the two proteins act in concert (van Dijk et al., 2005; Ishino, Chinzei and Yuda, 2005; van Schaijk et al., 2008; Labaied et al., 2007). […] Whether B9, P38 and P12p associate with P52 and/or P36 and contribute to sporozoite invasion still deserves further investigations.”

Sequestrin is not expressed at the sporozoite stage, but only later during liver stage maturation (Orito et al. Mol Microbiol 2013, PMID 23216750). This is clarified in the revised manuscript: “As mentioned above, sequestrin was reported to interact with CD36 (Ockenhouse et al., 1991), although the functional relevance of this interaction remains to be determined, as sequestrin is only expressed towards the end of liver stage development (Orito et al., 2013)”.

*P. falciparum* mutants deficient for P52 and P36 have been described, all showing a defect in hepatocyte infection. The studies describing these mutants are now cited in our revised manuscript as references (van Schaijk et al., 2008; vanBuskirk et al., 2009 and Mikolajczak et al., 2014. Further dissection of the role of P36 in *P. falciparum* implies determining whether changing its sequence alters host receptor usage in transgenic *P. falciparum* sporozoites. Such an approach would benefit from efficient genome editing methods recently developed in this species, such asthe CRISPR/Cas9 technology. However, our experiments using *P. falciparum* and *P. vivax* genes expressed in transgenic *P. berghei* (see point #2 above) illustrate that such an approach may not be straightforward. For these reasons, we believe that investigations based on *P. falciparum* mutants first require the identification of the other factors that cooperate with P36 to mediate sporozoite entry, and thus fall out of the scope of the present study.

*4) It has been shown by Mikolaijczak et al. in Molecular Therapy 2014 that double knockout P52/P36 parasites still invade the hepatocytes. How will the authors explain that P36 is the major determinant in the light of these even in P. Falciparum? Introducing at least some discussion on these results will strengthen the manuscript.*

Indeed, two studies have shown that *P. falciparum P52/P36* double knockout sporozoites still invade HC-04 hepatoma cells (VanBuskirk et al. PNAS 2009, PMID 19625622, and Mikolajczak et al. Molecular Therapy 2014, PMID 24827907). These studies are now cited in the revised text as vanBuskirk et al., 2009 and Mikolajczak et al., 2014. However, these studies analyzed *P. falciparum* sporozoite invasion without distinguishing between cell traversal and productive invasion events. *P. falciparum* sporozoites show robust cell traversal activity yet very low levels of productive infection in HC-04 cells (see Dumoulin et al. PLoS One 2015, PMID 26070149). Therefore, a vast majority of intracellular sporozoites likely correspond to non-productive invasion events associated with cell traversal, with both WT and KO parasites, as P52/P36 deficient parasites display no defect in cell traversal activity.

This point is now clarified in our revised manuscript: “It should be noted that standard invasion assays, as performed in these studies, do not distinguish between sporozoite productive entry and non-productive invasion events associated with cell traversal, complicating the interpretation of phenotypic analysis of the mutants. Here, using GFP-expressing *p52* and *p36* mutants and a robust FACS-based invasion assay (Risco-Castillo et al., 2015), we unequivocally establish that *P. yoelii* and *P. berghei* sporozoites lacking P52 and P36 efficiently migrate through cells but do not commit to productive invasion, reproducing the phenotype observed upon blockage of CD81 or SR-BI.”